# Structural basis of TRPC4 regulation by calmodulin and pharmacological agents

Deivanayagabarathy Vinayagam[1], Dennis Quentin[1], Jing Yu-Strzelczyk[2], Oleg Sitsel[1], Felipe Merino[1†], Markus Stabrin[1], Oliver Hofnagel[1], Maolin Yu[3], Mark W Ledeboer[3], Georg Nagel[2], Goran Malojcic[3], Stefan Raunser[1]*

[1]Department of Structural Biochemistry, Max Planck Institute of Molecular Physiology, Dortmund, Germany; [2]Department of Neurophysiology, Physiological Institute, Julius-Maximilians-Universität Würzburg, Würzburg, Germany; [3]Goldfinch Bio, Cambridge, United States

**Abstract** Canonical transient receptor potential channels (TRPC) are involved in receptor-operated and/or store-operated $Ca^{2+}$ signaling. Inhibition of TRPCs by small molecules was shown to be promising in treating renal diseases. In cells, the channels are regulated by calmodulin (CaM). Molecular details of both CaM and drug binding have remained elusive so far. Here, we report structures of TRPC4 in complex with three pyridazinone-based inhibitors and CaM. The structures reveal that all the inhibitors bind to the same cavity of the voltage-sensing-like domain and allow us to describe how structural changes from the ligand-binding site can be transmitted to the central ion-conducting pore of TRPC4. CaM binds to the rib helix of TRPC4, which results in the ordering of a previously disordered region, fixing the channel in its closed conformation. This represents a novel CaM-induced regulatory mechanism of canonical TRP channels.

*For correspondence:
stefan.raunser@mpi-dortmund.
mpg.de

Present address: †Department of Protein Evolution, Max Planck Institute for Developmental Biology, Tübingen, Germany

## Introduction

Transient receptor potential (TRP) ion channels mediate a plethora of vital cellular functions, including nociception, mechanosensation and store-operated $Ca^{2+}$ signaling (*Clapham, 2003*). Members belonging to the canonical TRP subfamily (TRPC) are involved in neuronal development and plasticity, as well as in vasorelaxation and kidney dysfunction (*Hall et al., 2019*; *Kochukov et al., 2013*; *Phelan et al., 2013*; *Riccio et al., 2009*). As a result, malfunction is often linked to pathologies such as neurological disorders and cardiac hypertrophy (*Selvaraj et al., 2010*; *Wu et al., 2010*). This class of non-selective cation channels can be further subdivided into TRPC1/4/5, TRPC2 (which is a pseudogene in humans) and TRPC3/6/7 groups, based on their sequence similarity. Within the TRPC1/4/5 sub-group, TRPC4 and TRPC5 share the highest sequence identity of 70% (*Plant and Schaefer, 2003*). Both proteins can form homo-tetrameric channels that allow the passage of $Ca^{2+}$, but also $Na^+$ ions to a lesser extent (*Minard et al., 2018*; *Owsianik et al., 2006*). In contrast, TRPC1, which shares approximately 48% identity with TRPC4/5, rather participates in the formation of hetero-tetrameric TRPC1/4/5 channels (*Bröker-Lai et al., 2017*). Whether or not functional homo-tetrameric TRPC1 channels exist in vivo and what their potential physiological impact may be, is currently not known.

TRPC4 is widely expressed in various tissues associated with the nervous-, cardiovascular- and immune system (*Freichel et al., 2014*). It has been shown to be necessary for neurite outgrowth, and its expression is upregulated in axonal regeneration after nerve injury (*Wu et al., 2008*). Channel activation results in a depolarization of the cell membrane, followed by a surge of intracellular $Ca^{2+}$ levels. The regulation of the activity of TRPC channels, however, is multi-faceted and ranges from modulation by endogenous and dietary lipids to surface receptors, redox environment and various types of cations (*Jeon et al., 2012*). TRPC4 was shown to be regulated by the ER-resident calcium

sensor Stim1, the lipid-binding protein SESTD1 and the G-protein $G_{\alpha i2}$ (*Jeon et al., 2012*; *Lee et al., 2010*; *Miehe et al., 2010*; *Zeng et al., 2008*). Dependent on the respective cellular environment in combination with the experimental method used for the measurement, TRPC4 was proposed to either act as a receptor-operated channel (ROC) (*Schaefer et al., 2000*; *Schaefer et al., 2002*) or as a store-operated channel (SOC) (*Wang et al., 2004*; *Warnat et al., 1999*).

In the studies, describing TRPC4 as SOC, two key proteins, calmodulin (CaM) and inositol 1,4,5-triphosphate receptor ($IP_3R$), compete for the same binding site on TRPC4 (*Mery et al., 2001*; *Tang et al., 2001*). First, $Ca^{2+}$-dependent binding of CaM inhibits the channel in the resting state. When intracellular $Ca^{2+}$ levels decrease, the activating $IP_3$ receptor directly interacts with TRPC4, displacing CaM, to restore channel activity (*Kanki et al., 2001*). This process, which is also known as conformational coupling, represents a primary regulation mechanism of gating for SOCs (*Berridge, 2004*). However, a detailed mechanistic understanding of CaM inhibition or $IP_3R$ activation remains elusive.

As ROC, the channel is activated by $G_{\alpha i2}$ coupled to the phospholipase C (PLC) signaling pathway (*Schaefer et al., 2000*). Specifically, the secondary messenger DAG, which is released upon the hydrolysis of phosphatidylinositol-4,5-bisphosphate (PIP2) by PLC, directly binds to and activates TRPC4 and TRPC5 (*Mederos y Schnitzler et al., 2018*).

Due to their implication in various diseases, TRPC channels also constitute a prime target for pharmacological intervention by small molecules (*Minard et al., 2018*). Activation of channels by the natural compound (-)-Englerin A (EA), which shows high potency and selectivity for TRPC4/5, inhibits tumor growth of renal cancer cells through increased $Ca^{2+}$ influx (*Akbulut et al., 2015*; *Carson et al., 2015*). Other activators include riluzole, BTD and the glucocorticoid methylprednisolone (*Beckmann et al., 2017*; *Richter et al., 2014*). However, these compounds are typically either less potent or show varying specificity.

Inhibitors of TRPC4/5 are mostly used to target renal diseases such as focal segmental glomerulosclerosis (FSGS) (*Mundel et al., 2019*; *Zhou et al., 2017*), but can also have a therapeutic effect on the central nervous system (CNS) (*Just et al., 2018*; *Yang et al., 2015*). Currently, two compounds are in clinical trials aiming to treat a proteinuric kidney disease and anxiety disorder/depression, conducted by Goldfinch Bio (NCT03970122) and Hydra/Boehringer Ingelheim (NCT03210272), respectively (*Mundel et al., 2019*; *Wulff et al., 2019*). While xanthine-based inhibitors, such as HC-070 and HC-608 (formerly known as Pico145) have assisted in advancing the field of pharmacological modulation of TRPC1/4/5 due to their exceptional high potency, they suffer from poor physiochemical properties such as low solubility (*Just et al., 2018*; *Rubaiy et al., 2017*).

Recently, a novel class of small molecule modulators selective for TRPC4/5 was identified in a high-throughput screen, building up on a piperazinone/pyridazinone scaffold (*Yu et al., 2019*). Among this class of modulators are the inhibitors GFB-9289, GFB-8749 and the GFB-8438. In particular, GFB-8438 showed promise as a potential drug for the treatment of proteinuric kidney disease, exhibiting overall favorable in vitro and in vivo properties (*Yu et al., 2019*). In vitro, mouse podocytes were protected from protamine-induced injury when treated with the inhibitor. Importantly, GFB-8438 also demonstrated robust efficacy in a hypertensive deoxycorticosterone acetate (DOCA)-salt rat model of FSGS, in which both albumin concentration and total protein levels were significantly reduced (*Yu et al., 2019*). However, information on the TRPC4/5 binding site and the mode-of-action of this novel compound class are still lacking. The structures of TRPC6 in complex with the activator AM-0883 and the inhibitor AM-1473 have been reported recently (*Bai et al., 2020*). Following this study, the structures of TRPC5 in complex with the inhibitors clemizole, HC-070 and Pico145 have been reported by two different groups (*Song et al., 2020*; *Wright et al., 2020*). These structures show that the small molecules bind at the voltage-sensor-like (VSL) domain or at the lipid-binding pocket situated close to the pore region. Although insights gained from different apo structures of TRPC4 have advanced our understanding of this medically important TRPC subfamily (*Duan et al., 2018*; *Vinayagam et al., 2018*), a molecular understanding of pharmacological modulation by pyridazinone-based small molecules as well as key regulatory proteins such as CaM and $IP_3R$ remains unknown. Here we report five cryo-EM structures of TRPC4 in its apo form and in complex with the inhibitors GFB-8438, GFB-9289, and GFB-8749, as well as CaM. Based on the analysis of the structures we propose mechanistic pathways by which CaM and small molecules exert their action to modulate the activity of the channel.

## Results and discussion

### Cryo-EM structures of full-length TRPC4 in complex with small-molecule inhibitors

We previously reported the high-resolution apo structure of zebrafish TRPC4 in its closed state (*Vinayagam et al., 2018*). To understand how channel activity is modulated by pharmacological compounds, we examined the complex of TRPC4 with the inhibitors GFB-8438, GFB-9289, and GFB-8749. All three compounds belong to the same novel class of TRPC4/5-selective modulators, which contain a common piperazinone/pyridazinone core.

To determine the electrophysiological effect of the compounds on TRPC4, we performed two-electrode voltage clamp experiments with TRPC4-expressing *Xenopus* oocytes. Expectedly, perfusion of the oocytes with the known activator (-)-Englerin A (0.1 or 1 μM) induced large inward currents at –40 mV. However, perfusing the oocytes with increasing doses of GFB-9289, GFB-8438 or GFB-8749 did not induce any current change, even in inside-out excised patches from oocytes or in TRPC4-expressing HEK293 cells (data not shown). Instead, GFB-9289, GFB-8438 and GFB-8749 inhibited (-)-Englerin A activation in a dose-dependent manner (*Figure 1A–C,E–G and I–K*). Moreover, competition assays clearly showed that (-)-Englerin A and the respective compounds are competitive antagonists, although we cannot determine if they compete for the same binding site (*Figure 1D,H and L*).

We then formed the TRPC4 complexes with the respective small molecules. We did not add cholesteryl hemisuccinate and exogenous lipid molecules during purification to exclude potential interference with ligand binding. Using cryogenic electron microscopy (cryo-EM) and single particle analysis, we have determined the structures of GFB-8438, GFB-9289 and GFB-8749-bound TRPC4 to an average resolution of 3.6 Å, 3.2 Å, and 3.8 Å respectively (*Figure 2*, *Figure 2—figure supplements 1–3*).

Overall, the structures of these complexes are similar to the previously determined TRPC4 apo structure (*Figure 2A*, *Figure 2—figure supplements 2* and *4*). The architecture is typical for the canonical TRP channel family with a transmembrane region (TM domain) where the pore region of one protomer domain-swaps with the voltage-sensor-like (VSL) domain of another. The cytosolic domain harboring the ankyrin repeat (AR) embraces the coiled coil helix in the center and the N-terminal ankyrin domain is associated with the TM domain via a helical linker domain. The C-terminal helix connects to the TM domain through the rib and TRP helix which also bridges the TM domain with the helical linker domain (*Figure 2—figure supplement 5*).

### TRPC4 in complex with inhibitor GFB-8438

In the GFB-8438-bound structure, we found an additional density compared to the apo structure inside a cavity formed by the VSL domain, TRP helix and re-entrant loop (*Figure 2D*). The shape of the density clearly indicated that it corresponds to the bound inhibitor. The shape and the surrounding chemical environment allowed us to build the model of the inhibitor inside this extra density (*Figure 2D*). Notably, the inhibitor AM-1473, which belongs to a different class of small molecules, was shown to bind to a similar region in TRPC6 (*Bai et al., 2020*).

The chemical structure of GFB-8438 consists of three six-membered rings: a pyridazinone ring and trifluoromethyl benzyl group at opposing ends are connected by a central 1,4-disubstituted piperazinone ring (*Figure 2D*). Its binding to the protein is predominantly mediated by hydrophobic contacts (*Figure 2D*). The nitrogen, the chlorine, and the oxo group of the pyridazinone ring form hydrogen bonds as well as halogen bonds with N442 of helix S3, Y373 of helix S1 and S488 of the S4 helix, respectively. The hydrophobic part of the pyridazinone ring is stabilized by a π-π stacking interaction with F413 of helix S2 on one side and M441 of helix S3 on the opposite side. The middle piperazinone ring forms a hydrophobic interaction with the Y373 while the oxo-group of the ring is engaged in a hydrogen bond with R491 of the S4 helix. The tri-fluoromethyl benzyl group engages in a hydrophobic interaction with L495 of helix S4, and the fluoride group is involved in a hydrogen bond with H369 of S1 and Y646 from the TRP helix. The residues interacting with the inhibitor are identical between TRPC4 and TRPC5 (*Figure 3—figure supplement 1*) indicating a similar ligand binding mode in TRPC5, which is supported by their close $IC_{50}$ values of 0.18 and 0.29 μM for human TRPC5 and human TRPC4, respectively (*Yu et al., 2019*).

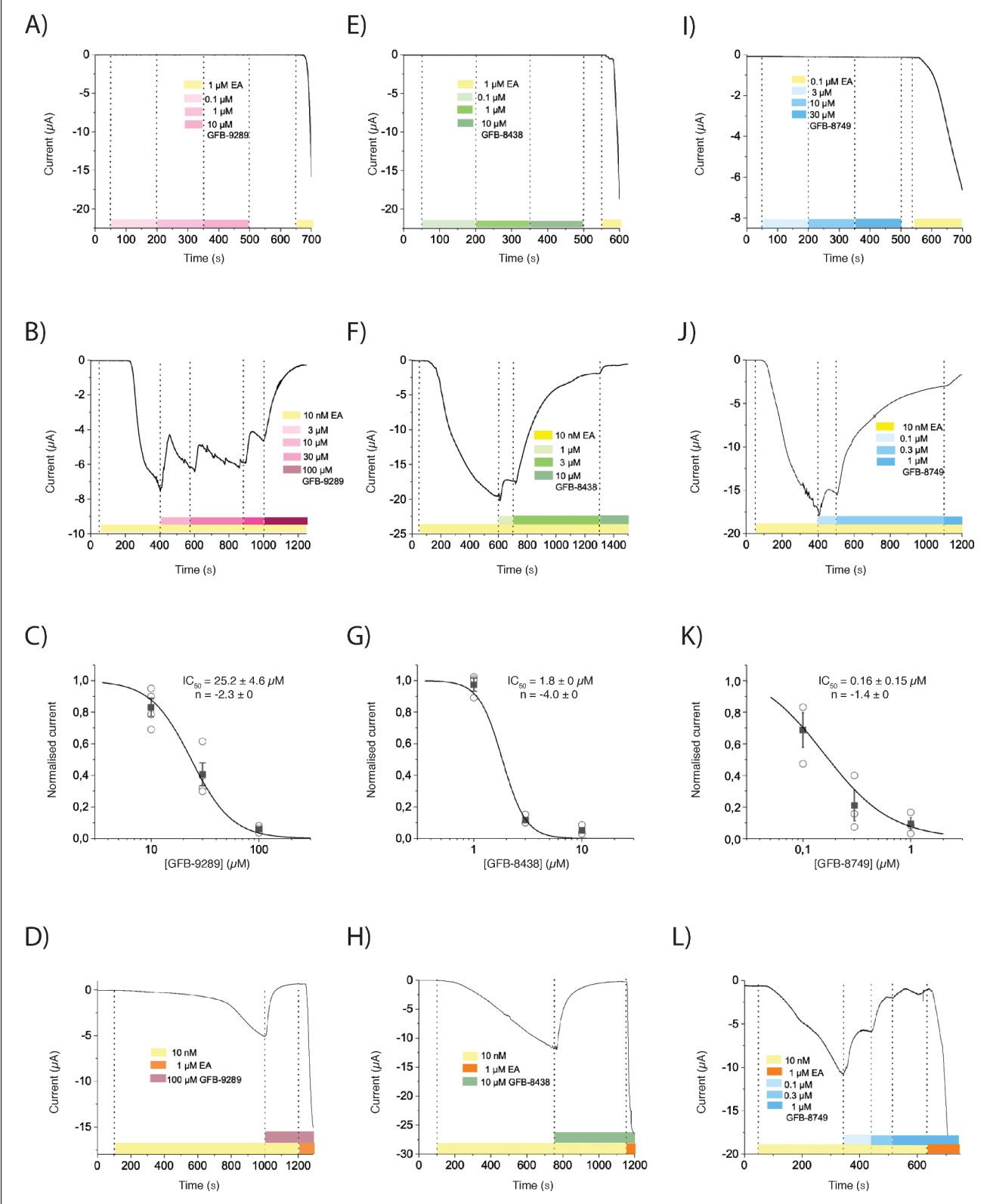

**Figure 1.** The effect of three pyridazinone-based small molecules on TRPC4 and on (-)-Englerin A (EA)-induced opening of TRPC4. (**A, E and I**) TRPC4-expressing *Xenopus* oocytes were held at −40 mV and perfused with increasing concentrations of GFB-9289 (**A**), GFB-8438 (**E**) or GFB-8749 (**I**) to test the potential activation effect. 0.1 or 1 μM (-)-Englerin A (EA) were given at the end of recording to confirm the functional expression of TRPC4. (**B, F and J**) After activation of TRPC4 with 10 nM (-)-Englerin A, various concentrations of GFB-9289 (**B**), GFB-8438 (**F**) or GFB-8749 (**J**) were given together

*Figure 1 continued on next page*

*Figure 1 continued*

with 10 nM (-)-Englerin A to test the inhibitory effect. (**C, G and K**) Hill equation: y = 1/(1+ (IC$_{50}$/x)^n) was used to fit the dose-inhibition curve, where IC$_{50}$ is the 50% inhibitory concentration, x is the concentration of the ligand and n is the Hill coefficient. (**D, H, and L**) Sufficient GFB-9289 (**D**), GFB-8438 (**H**) or GFB-8749 (**L**) were used to fully inhibit 10 nM Englerin A-induced inward current, which can be reactivated by 1 µM Englerin A. Symbols with error bars represent mean ± SEM (n ≥ 3). The colored bars between the dashed lines indicate the concentration of compounds which was kept constant over time.

## TRPC4 in complex with inhibitor GFB-9289

As in the GFB-8438 inhibitor-bound structure, we found an extra density inside the VSL domain region of GFB-9289-bound TRPC4 (*Figure 2E*, *Figure 2—figure supplement 2A and B*). In addition to the surrounding chemical environment, the high resolution of the map enabled us to unambiguously build the ligand (*Figure 2E*).

Similar to GFB-8438, the chemical structure of GFB-9289 consists of three six-membered rings: a pyridazinone ring and a cyclohexyl group at opposing ends are connected by a central 1,4-disubstituted piperazinone ring (*Figure 2E*). The key difference between the molecules is the terminal ring, which is a tri-fluoromethyl benzyl group in the case of GFB-8438 and a cyclohexyl ring in GFB-9289. Given the common chemical scaffold structure with the difference limited to one part of the molecule, it is not surprising that they bind to the same region. This suggests that the VSL domain is a highly sensitive regulatory domain that responds to subtle stimuli in its small ligand-binding pocket in order to govern the function of this large tetrameric macromolecular complex.

Most interactions of GFB-9289 with the S1-S4 helices are the same as they are for GFB-8438 with small residue movements to accommodate the slightly different structure of GFB-9289 (*Figure 3A–C*). In the case of GFB-9289, Y373 forms hydrophobic interactions with the cyclohexyl and piperazinone rings. A reconfiguration of the binding interactions to the pyridazinone ring now includes hydrogen bonds to S488 side chain via its oxygen atom, unlike the halogen bond with Y373 observed in the inhibitor complex described above. The reduced size of the cyclohexyl group results in the reorientation of interacting residues. Importantly, Y646 of the TRP helix and H369 of the S2 helix do not interact with the compound and are rotating away from the interface (*Figure 3B C*). This reduced stabilization of GFB-9289 is the likely cause for the weaker binding in comparison to the GFB-8438 (*Figure 1*).

## TRPC4 in complex with inhibitor GFB-8749

Similar to the GFB-8438 and GFB-9289 inhibitor-bound structure, an extra density was observed inside the VSL domain region of GFB-8749-bound TRPC4 (*Figure 2F*, *Figure 2—figure supplement 2C and D*). The shape, the surrounding chemical environment along with structural similarity of the ligand with GFB-8438 and GFB-9289 guided us to unambiguously build the ligand inside the density (*Figure 2F*).

The structure of GFB-8749 shares a common scaffold with GFB-8438 and GFB-9289, consisting of three six-membered rings. A cyclohexyl group and a pyridazinone ring at opposing ends are connected by a central 1,4-disubstituted piperazinone ring (*Supplementary file 1*). The main difference between GFB-8749 and the other compounds is in the terminal cyclohexyl ring with a substitution of a difluoro group at the para position (*Figure 2D–F*). Thus, the interaction of the protein residues with the pyridazinone ring and 1,4-disubstituted piperazinone ring are found to be almost identical with that of GFB-8438 and GFB-9289 (*Figure 2D–F*). The terminal cyclohexyl ring substituted with the difluoro group is stabilized by the hydrogen bond interaction with Y646 of the TRP helix and the cyclohexyl ring is stabilized by hydrophobic interaction with L495 of helix S4 and Y373 of helix S1 (*Figure 3D*).

Given the similar structure of all the three compounds, it is intriguing to suggest that all pyridazinone-based compounds target the VSL domain of TRPC4/5 to modulate the activity of the channel.

## Structural rearrangements in the ligand-binding pocket

To understand the structural rearrangement upon ligand binding, we compared the ligand-bound structures with the structure of TRPC4 in the apo state (*Vinayagam et al., 2018*; *Figure 3A–D*). In the apo structure, some of the residues of the ligand binding pocket interact with each other via

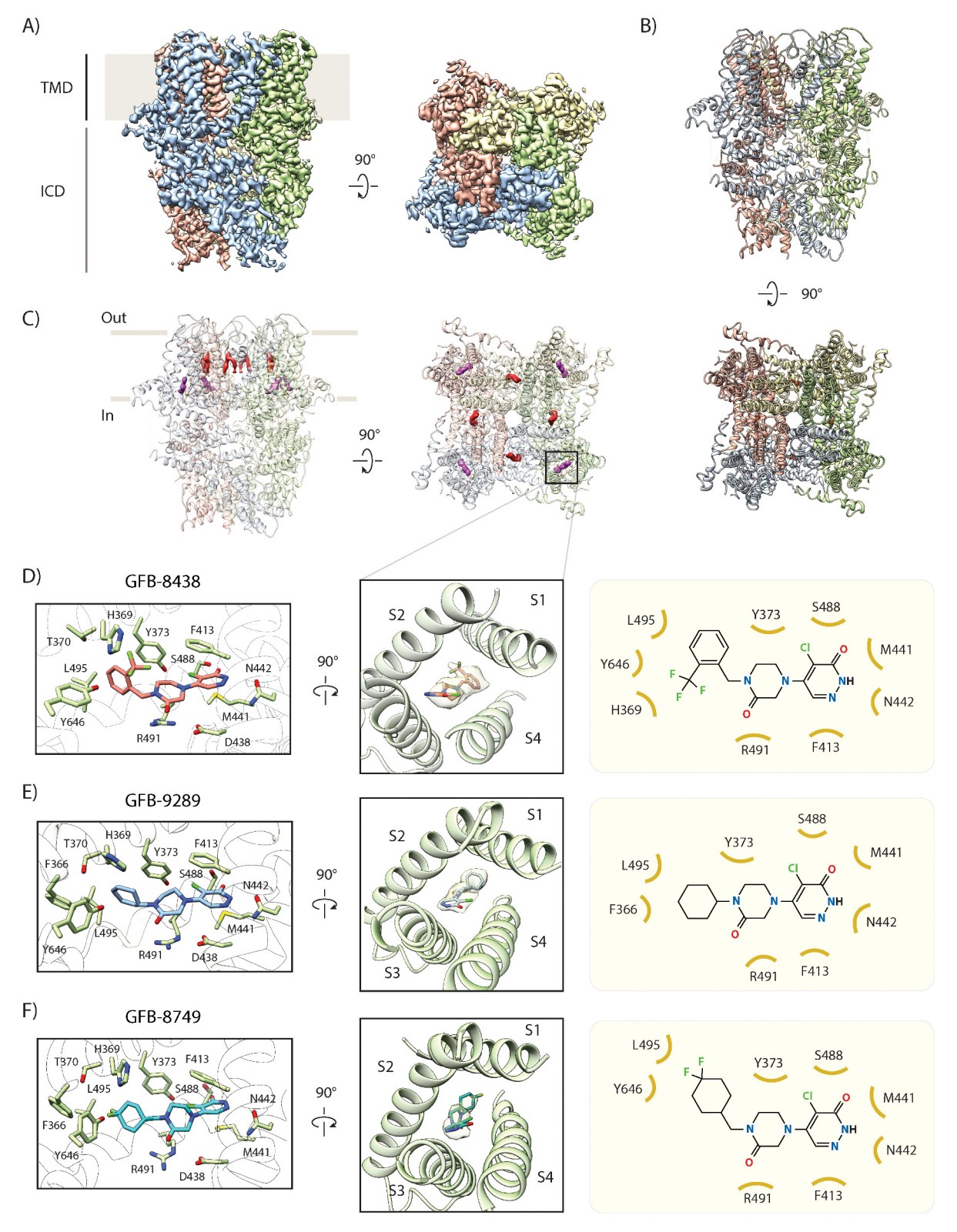

**Figure 2.** Cryo-EM structure of inhibitor-bound TRPC4 channel. (**A**) Side and top view of the cryo-EM map of GFB-8438 inhibitor-bound TRPC4, with each subunit colored differently. Positions of the transmembrane domain (TMD) and intracellular cytosolic domain (ICD) are indicated. (**B**) Side and top view of the structure of GFB-8438 inhibitor-bound TRPC4, with each subunit colored differently. (**C**) Location of non-protein densities relative to the atomic model of TRPC4, which is shown in transparent ribbon representation in the side- and top view. Densities corresponding to lipids are depicted

*Figure 2 continued on next page*

*Figure 2 continued*

in red, GFB-8438 density is shown in purple. (D) Close-up of the ligand-binding pocket, showing the density corresponding to the inhibitor GFB-8438 (transparent) with the ligand structure modelled inside. GFB-8438 is enclosed by the four helices S1 to S4, constituting the VSL domain. A rotated view of the ligand-binding pocket is shown in the left panel with important and interacting residues highlighted. GFB-8438 is shown in purple. In the right panel the chemical structure of the TRPC4 inhibitor GFB-8438 is shown, with important and interacting residues of TRPC4 highlighted. Non-carbon atoms are colored according to element, with halogens in green, nitrogen in blue and oxygen in red. (E) and (F) Same in (D) but for inhibitor GFB-9289 and GFB-8749 bound structures of TRPC4 respectively.

The online version of this article includes the following figure supplement(s) for figure 2:

**Figure supplement 1.** Cryo-EM image processing workflow for TRPC4 in complex with the inhibitor.
**Figure supplement 2.** Cryo-EM map of TRPC4 in complex with the inhibitor GFB-9289 and GFB-8749.
**Figure supplement 3.** Local resolution of the TRPC4-ligand complex maps.
**Figure supplement 4.** Comparison between different TRPC4 structures.
**Figure supplement 5.** Domain architecture of zebrafish TRPC4 channel.

hydrophobic (Y373, F413, M441) and hydrophilic interactions (R491 and E438) (*Figure 3A*). Upon ligand binding, these residues move and reshape the pocket to accommodate the ligands, indicating an induced-fit mechanism or conformational selection (*Hammes et al., 2009*; *Figure 3B–D*). Similarly, the side chains of L495 and H369 rotate, move or flip to accommodate and stabilize the interaction with the cyclohexyl group or the benzyl group (*Figure 3B–D*). These ligand-specific arrangements of the ligand binding pocket highlight its plasticity.

The inhibitor GFB-8438 has been shown to be more specific for TRPC4/5 than TRPC6 (*Yu et al., 2019*). Comparison of the TRPC4/5 binding pocket with TRPC6 reveals a critical difference in ligand binding residues (*Figure 3E*). The cognate N442 residue in TRPC4/5 is replaced by L534 in TRPC6, which abrogates hydrogen bond formation with the nitrogen atom of pyridazinone. F413, which in TRPC4 forms a π-π interaction with the pyridazinone ring, is replaced by the weakly interacting hydrophobic residue M505 in TRPC6 (*Figure 3E F*). These crucial substitutions in the ligand binding pocket explain the much lower binding affinity of TRPC6 for GFB-8438 (>30 μM) (*Yu et al., 2019*).

Interestingly, we have observed that the inhibitors GFB-8438 and GFB-8749, which have lower $IC_{50}$ values than GFB-9289, interact with both the TRP helix and the VSL domain, whereas GFB-9289 exclusively interacts with the VSL domain (*Figures 1*, *3G*). This suggests that interaction of the compounds with the peripheral VSL domain is mainly responsible for the allosteric inhibition of the channel, and additional interaction of the inhibitor with the TRP helix amplifies this effect. We hypothesize that the direct stabilizing interaction with the TRP helix constrains it and the adjacent S6 helix thereby arresting the channel in a closed state. In this manner, the allosteric interactions within a peripheral binding site propagate to the ion pore in the center of the protein.

Inhibitors were also shown to bind to the VSL domain in other TRP family members (*Bai et al., 2020*; *Diver et al., 2019*; *Singh et al., 2018b*), indicating that this domain acts in general as a regulatory region in TRP channels (*Figure 3G*). Interestingly, besides GFB-9289, all VSL-bound inhibitors not only interact with the VSL domain but also with the TRP helix.

## Ligand-induced changes in TRPC4

Besides the structural rearrangements in the ligand binding pockets, we did not observe major ligand-induced conformational changes in TRPC4. Similar to the apo structure, the channel is closed at the lower gate in all structures of the inhibitor-bound TRPC4. The lower gate shows a minimal constriction defined by residue N621 with a van der Waals surface diameter of approximately 0.7 Å, which is too narrow for $Ca^{2+}$ to pass through (*Figure 4A–B*).

We found small differences in the selectivity filter. Surprisingly, the backbone residues F576 and G577 forming the TRPC4 selectivity filter show a slightly wider radius in the ligand-bound structures compared to the apo structure. This could be due to a density in the selectivity filter which we did not observe in the apo structure, indicating that a cation, presumably $Ca^{2+}$ or $Na^+$, resides in the filter, while the filter is empty in the apo form (*Figure 4A*).

In the ligand-bound structures a characteristic lipid density is situated close to the pore which is either phosphatidic acid or ceramide-1-phosphate, its structural analogue (*Vinayagam et al., 2018*). Since we did not add lipids during our purification, this annular lipid likely co-purified with the

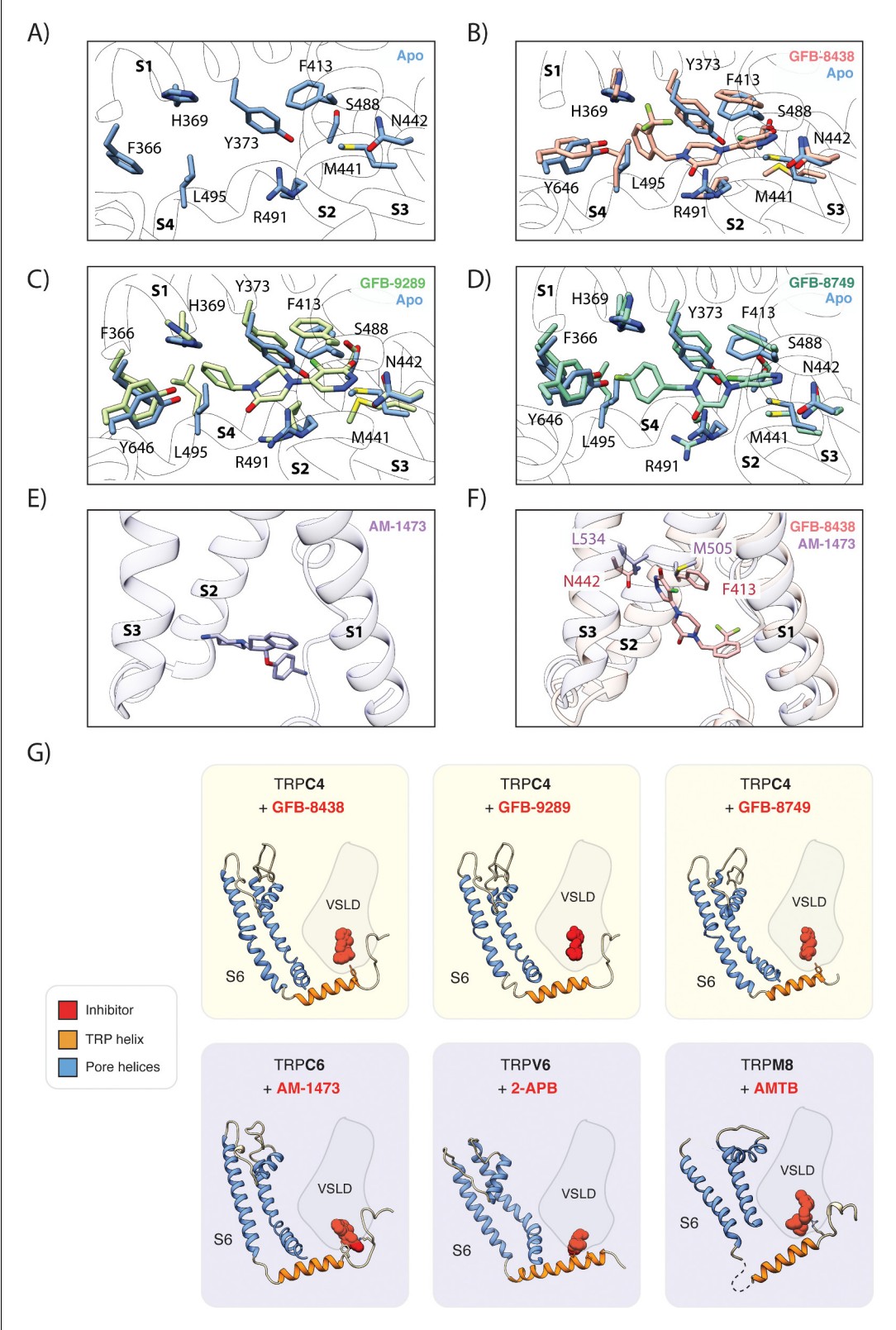

**Figure 3.** Comparison of the ligand-binding pocket in TRPC4. (**A**) Close-up of ligand-binding pocket in the apo TRPC4 structure, which is enclosed by the four helices S1 to S4 of the voltage sensing-like domain. (**B**) Superposition of inhibitor-bound (red) and apo (blue) structure of TRPC4. A close-up of the ligand-binding pocket is shown, with important and interacting residues highlighted. The inhibitor GFB-8438 is depicted in red, positions of the surrounding helices S1 to S4 are indicated. (**C**) and (**D**) Same as in (**B**) for the inhibitor GFB-9289 and GFB-8749 -bound TRPC4 structures respectively.

*Figure 3 continued on next page*

*Figure 3 continued*

The structures of GFB-9289 and GFB-8749 are depicted in green and cyan respectively. In all the inhibitor-bound structures, several residues move away from the center of the pocket to create space for accommodating the respective ligand. (**E**) Position of the inhibitor AM-1473 within the VSL domain binding pocket of TRPC6 is shown. The surrounding helices S1-S3 are indicated for orientation. (**F**) Superposition of GFB-8438-bound TRPC4 (red) and AM-1473-bound TRPC6 (purple) channels. The location of the GFB-8438 inhibitor within the VSL domain is shown. In contrast to AM-1473, which is located in the lower part of the binding pocket (see E), GFB-8438 additionally interacts with the upper region of the pocket. The depicted residues in this region contribute to the selectivity of GFB-8438 for TRP4/5 channels. (**G**) Comparison of small-molecule modulators of the TRP channel family that target the ligand-binding pocket enclosed by the helices of the VSL domain (VSLD). Small molecules are depicted as space-filled spheres with inhibitors shown in red. Residues interacting with the ligand are shown in stick representation. Pore helices are colored in blue, the TRP helix in orange.

The online version of this article includes the following figure supplement(s) for figure 3:

**Figure supplement 1.** Sequence alignment of zebrafish TRPC4, human TRPC4, TRPC5, and TRPC6.

protein. Each of the two lipid tails is placed like an anchor between neighboring S5 and S6 helices by forming several hydrophobic interactions (*Figure 4—figure supplement 1*). We hypothesize that this lipid site could be crucial for the gating of the channel, since small molecules can bind in this region, and modulate the channel as observed in activator-bound TRPC6 (*Bai et al., 2020*).

We identified density for a putative cation in the $Ca^{2+}$ binding site of the VSL domain in the ligand-bound structures (*Figure 4C*, *Figure 4—figure supplement 2*), also present in the apo structure of TRPC4 (*Vinayagam et al., 2018*). The ion binding site is coordinated by the carbonyl oxygen of D438 and N435 of the S3 helix, along with E417 and Q420 of the S2 helix, which are the favorable coordination residues for an alkaline earth metal ion such as $Ca^{2+}$ (*Zheng et al., 2017a*). Interestingly, not only is a hydroxyl group of Y429 close to the density but also an oxo group of the ligands,

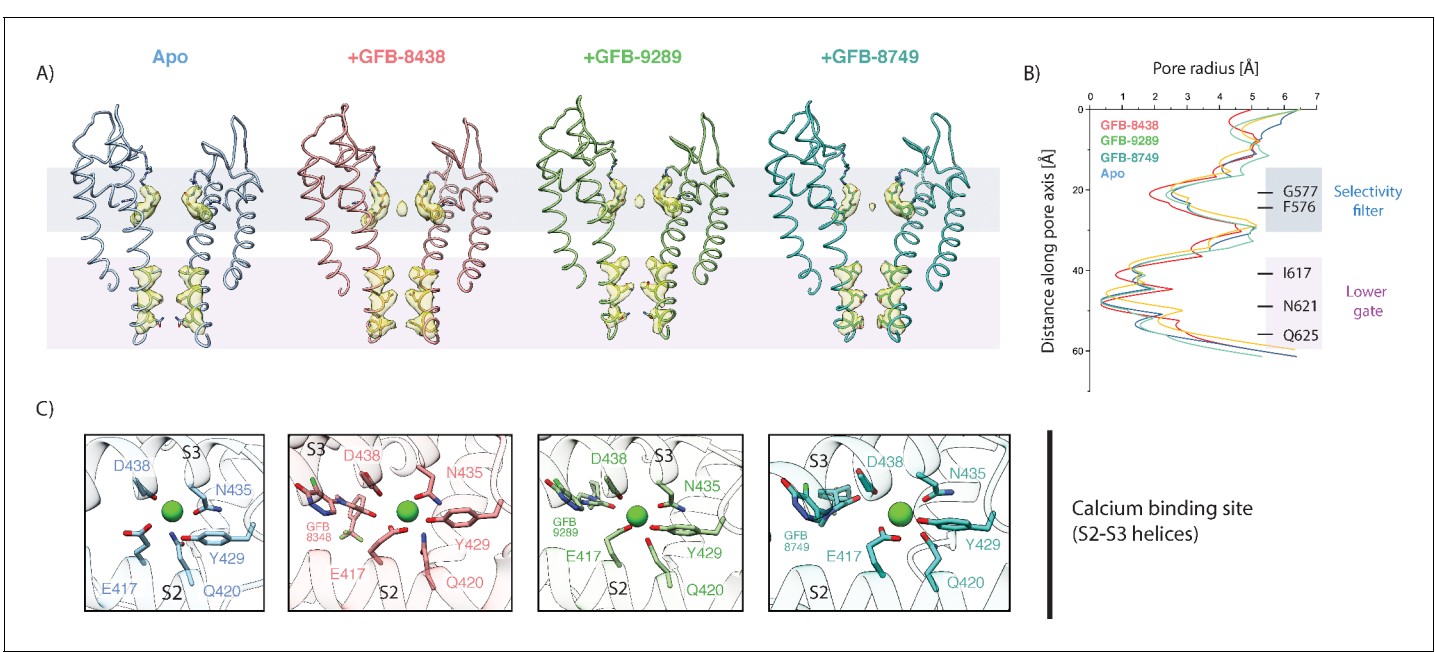

**Figure 4.** Comparison of the ion conduction pore and $Ca^{2+}$-binding site. (**A**) Side view of the pore-forming region of TRPC4 in the apo- (blue), GFB-8438 (red) GFB-9289 (green) and GFB-8749 (cyan-blue) inhibitor-bound structures. Only the two opposing subunits of the tetrameric channel are shown as ribbon representation for clarity. The density at comparable thresholds corresponding to the selectivity filter (light blue) and the lower gate (pink) is shown. A central density is observed in all maps, except the apo structure. (**B**) The calculated pore-radii corresponding to the four TRPC structures in (**A**) are depicted. The color code is also identical to (**A**). The positions of important residues, constituting the selectivity filter and the lower gate, are indicated on the right. (**C**) Close-up of the $Ca^{2+}$-binding site in the four TRPC4 structures, located in direct vicinity to the ligand binding pocket of the VSL domain. Position of ligands and coordinating residues are highlighted. Color code of TRPC4 structures is as in (**A**).

The online version of this article includes the following figure supplement(s) for figure 4:

**Figure supplement 1.** Different views of the lipid binding pocket at the interface between two subunits.

**Figure supplement 2.** $Ca^{2+}$-binding site in the VSL domain of apo and ligand-bound TRPC4.

which could complete the octahedral coordination of $Ca^{2+}$ via a bridging water molecule. The presence of the ligands could thus help to stabilize bound $Ca^{2+}$.

TRPC5 and TRPC4 activation has been reported to be $Ca^{2+}$-dependent (*Plant and Schaefer, 2003*). Similar to our observation here, the binding of $Ca^{2+}$ has been described for TRPM4 and TRMP8, both of which are also known to be activated by $Ca^{2+}$. The structures of these channels are in a closed conformation representing the desensitized state (*Autzen et al., 2018*; *Diver et al., 2019*). Considering this, the molecular role of the VSL domain-bound calcium ion in activation or desensitization of the TRPC4 channel is a compelling topic for further investigation.

## Structure of TRPC4 in complex with CaM

CaM has been shown to bind and regulate the TRPC4 channel (*Zhu, 2005*; *Tang et al., 2001*). At high $Ca^{2+}$ concentrations in the cytosol, CaM binds in its $Ca^{2+}$-bound state to TRPC4 and inhibits $Ca^{2+}$ entry. At low $Ca^{2+}$ concentrations, CaM changes its conformation and dissociates from the channel. The store-operated $Ca^{2+}$ entry pathway hypothesis (*Tang et al., 2001*) further proposes that CaM binding to the channel at resting state prevents TRPC4 from being spontaneously activated by $IP_3$ receptors. When $Ca^{2+}$ levels in the endoplasmic reticulum (ER) - but not in the cytosol - drop, the affinity of the $IP_3$ receptor to TRPC4 increases and CaM is displaced through a conformational coupling mechanism (*Rosado et al., 2015*; *Tang et al., 2001*). This activates the TRP channel. To further understand the mechanistic process of CaM inhibition, we set out to determine the structure of the TRPC4-CaM complex.

We first performed a pull-down experiment using a CaM Sepharose column at high $Ca^{2+}$ concentrations with CaM acting as bait to capture TRPC4. As expected, TRPC4 was trapped in the CaM column in presence of $Ca^{2+}$ and released by chelating the $Ca^{2+}$ with EGTA (*Figure 5—figure supplement 1*). This is in line with previous studies which used smaller peptides of TRPC4 instead of the full-length protein used in our experiment (*Tang et al., 2001*). Since $Ca^{2+}$ is necessary for the binding of CaM to TRPC4, we prepared the protein sample in the detergent LMNG (lauryl maltose neopentyl glycol) instead of following the amphipol exchange methodology that we used previously. Amphipols are known to interact with $Ca^{2+}$ ions and could thus disrupt CaM binding (*Le Bon et al., 2018*). We then determined the structure of TRPC4 in LMNG in complex with CaM. For the TRPC4-CaM complex, we added a 10-fold molar excess of CaM to tetrameric TRPC4 in the presence of 10 mM calcium chloride throughout the purification process after detergent extraction.

The CaM complex sample yielded a 3.3 Å map with applied C4 symmetry (*Figure 5—figure supplement 2*). We observed additional density surrounding the rib-helix termini protruding from the protein core, although the resolution in this region was lower than at the core of the protein (*Figure 5—figure supplement 3A*). Besides its localization at the periphery we suspected that an incomplete saturation of TRPC4 by CaM could be the reason for the lower local resolution. Hence, we performed 3D sorting without applied symmetry to resolve the subpopulations with different binding stoichiometries. 13% of the TRPC4 channels had one CaM-bound, 35% and 31% had two or three bound, respectively and only 20% were fully saturated (*Figure 5A*). In addition, some of the densities corresponding to CaM were less defined than others. The classes with clear CaM densities were then rotated and properly aligned (*Figure 5—figure supplement 2*). The final local resolution of CaM improved to a resolution of 4–5 Å (*Figure 5—figure supplement 3A*). We could clearly identify four helices that correspond to the helices of one lobe of CaM and flexibly fitted this part of the protein (*Figure 5B*). The other CaM lobe was not resolved, indicating that this part of the protein is more flexible in this complex.

Based on the flexibly fitted atomic model we observed that CaM not only binds to the tip of the rib helix (residues 691–703) and the following loop (residues 677–690) that connects the rib helix with a newly identified helix (residues 666–676), but it also interacts with the adjoining loop region comprising residues 273–277 (*Figure 5C*). The core region of CaM binds to TRPC4 by forming hydrophobic interactions, while the peripheral residues of CaM are stabilized by hydrophilic interactions (*Figure 5D*) that are typically observed in CaM-protein/peptide complexes (*Villalobo et al., 2018*).

The interacting residues of TRPC4 partially overlap with a peptide corresponding to residues 695–724 that have been previously shown to interact with CaM (*Tang et al., 2001*). Since our structure revealed that CaM only interacts with residues 688–703, we conclude that the residues 695–703

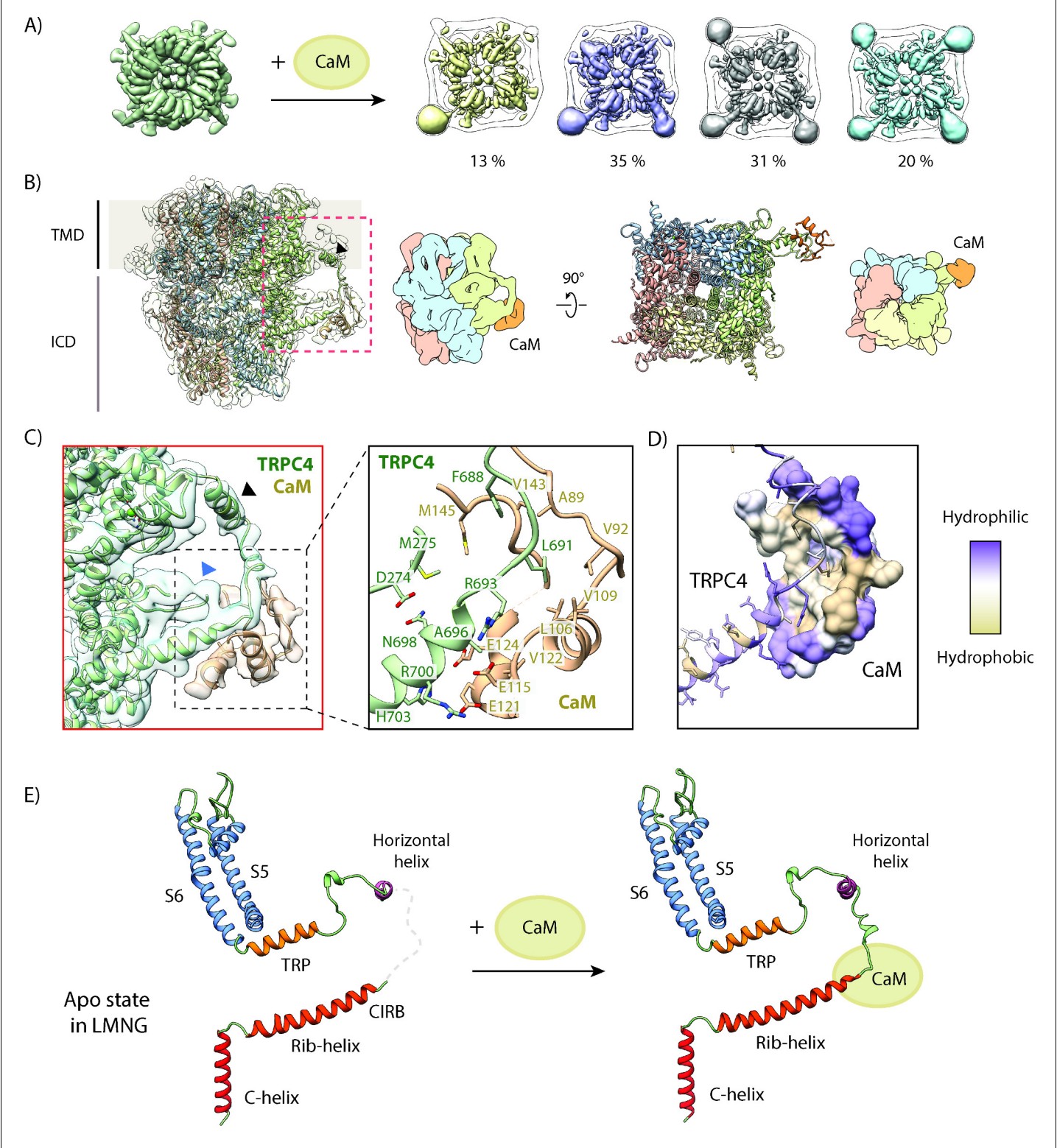

**Figure 5.** Structural basis for inhibition of TRPC4 by calmodulin. (A) One to four CaM molecules are bound to the CIRB binding sites of the tetrameric TRPC4 channel. 13% of particles are decorated with one (yellow), 35% with two (lilac), 31% with three (grey) and 20% with four CaM molecules (turquoise). (B) Side view of the CaM-bound TRPC4 density map (transparent) with the corresponding atomic model fitted inside, in which each protomer is colored differently. Position of the horizontal helix is indicated by black arrowhead. The bottom view of the atomic model is shown in the right panel. A schematic representation for both views is provided next to the atomic models. CaM is colored in orange. (C) Close-up of the indicated
*Figure 5 continued on next page*

*Figure 5 continued*

region in (B), showing the CaM binding region (left panel). CaM is colored in orange, TRPC4 in green. Positions of the horizontal helix and loop region 273–277 are indicated by black and blue arrowhead, respectively. Important and the predicted interacting residues of TRPC4 and CaM based on our model are highlighted in the right panel. (D) TRPC4 (cartoon representation) and CaM (surface representation) are colored according to hydrophobicity. There is a central hydrophobic cavity in CaM that is surrounded by hydrophilic residues in its periphery. The complementary binding region of TRPC matches this profile. (E) The C-terminal helix (red), the rib-helix (red-orange), the horizontal helix (purple), the TRP helix (orange) and the pore-forming helices (blue) of a single TRPC4 promoter are shown before (left panel) and after CaM binding (right panel). CaM binding stabilizes the previously disordered region connecting the rib-helix and horizontal/TRP-helix. LMNG – lauryl maltose neopentyl glycol.

The online version of this article includes the following figure supplement(s) for figure 5:

**Figure supplement 1.** Analysis of CaM binding to TRPC4 by biochemical methods.
**Figure supplement 2.** Cryo-EM image processing of the TRPC4- CaM complex.
**Figure supplement 3.** Local resolution maps of TRPC4-apo (LMNG) and TRPC4-CaM.
**Figure supplement 4.** Cryo-EM image processing and structure determination of TRPC4 solubilized in LMNG.
**Figure supplement 5.** Comparison of the ion conduction pore and $Ca^{2+}$-binding site.
**Figure supplement 6.** Biochemical and structural analysis of CaM N- and C-lobe binding to TRPC4.

are sufficient for CaM binding in vitro. Residues 704–725 of the rib helix interact with the protein core and are inaccessible for interaction with CaM.

## CaM-induced changes in TRPC4

To be able to identify CaM-induced structural effects, we also solved the structure of TRPC4 in its apo state under the same conditions as for the TRPC4-CaM complex without the addition of external lipids. The apo structure of TRPC4 in LMNG reached a resolution of 2.85 Å, allowing us to build an atomic model with high accuracy (*Figure 5—figure supplements 3B*, *4*). The overall structure is similar to the previously reported amphipol-exchanged apo structure of TRPC4 in the closed state (*Vinayagam et al., 2018*; *Figure 2—figure supplement 4*). However, in both the apo and CaM-bound structure, we observed for the first time an additional density corresponding to a horizontal helix located at the transmembrane-cytoplasmic interface outside the transmembrane core (residues 666–676) (*Figure 5B–D*, *Figure 5—figure supplement 4*). The hydrophobic residues of this helix face the transmembrane helix and the inner lipid leaflet while the hydrophilic residues project into the cytoplasm, giving the helix an amphipathic nature.

Comparing the TRPC4 apo structure with that of TRPC4-CaM, we could identify only small differences in the center of the channel. Both, the apo and CaM-bound TRPC4 structures showed the same constriction of 0.7 Å defined by N621 at the lower gate indicating the closed state of the channel (*Figure 5—figure supplement 5*). Interestingly, when CaM binds to TRPC4 the selectivity filter is slightly widened and contains a density that likely corresponds to $Ca^{2+}$ or $Na^+$ as in the case of the inhibitor-bound channel. We observed a much stronger density at the $Ca^{2+}$ binding site in the VSL domain for the CaM-bound structure compared to apo TRPC4, presumably due to the high $Ca^{2+}$ concentration we used for preparing the CaM-TRPC4 complex.

The differences between TRPC4-CaM and the TRPC4 apo structure are more pronounced at the periphery of the channel. There, CaM binding stabilizes a longer stretch (residues 677–692) of TRPC4 that is highly flexible in the apo state of the channel (*Figure 5B C*). Therefore, binding of CaM to this region of TRPC4 likely reduces the overall flexibility of the channel, fixing it in its closed state (*Figure 5E*). Since one TRPC4 tetramer can bind up to four CaMs, this suggests that the number of CaMs simultaneously bound to TRPC4 could fine tune the level of channel activity.

Importantly, this mechanism of CaM-mediated regulation completely differs from that described for other TRP channels, such as TRPV5 and TRPV6 (*Hughes et al., 2018*; *Singh et al., 2018a*). There, CaM binds in a 1:4 stoichiometry, with one CaM binding to the center of the tetrameric channels via the open cytoplasmic part, plugging it with its protruding lysine residue (*Figure 6*; *Hughes et al., 2018*; *Singh et al., 2018a*). In the TRPC4-CaM complex structure, the central core of the cytoplasmic region is occupied by a coiled coil helix. Thus, CaM cannot access the core of the cytoplasmic region in TRPC4. Other channels of the TRPC subfamily also contain this coiled coil helix and the rib helix (*Duan et al., 2018*; *Tang et al., 2018*). Therefore, we propose that the novel mechanism of CaM inhibition via binding to the rib helix is paradigmatic for all TRPCs.

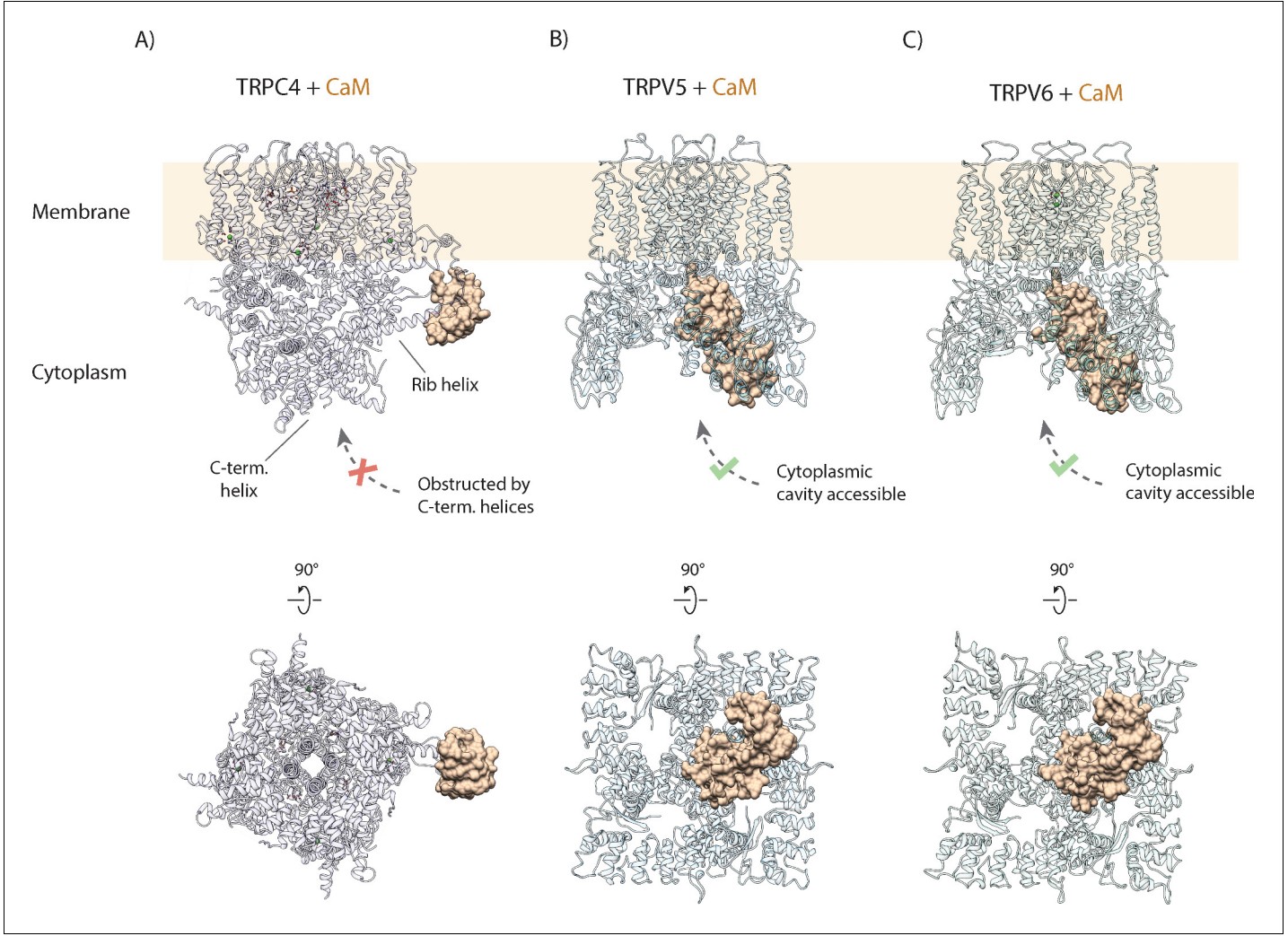

**Figure 6.** Comparison of CaM binding in TRPC and TRPV channels. (A) Calmodulin (CaM) interacts with the rib helix of TRPC4. Side (upper panel) and bottom (lower panel) view of the CaM-bound TRPC4 is shown, with TRPC4 structure in cartoon representation with moderate transparency and CaM in space filling sphere representation. Only a single lobe of the double-lobed CaM molecule is resolved in the structure. This indicates that the second lobe is rather flexible. Up to four binding sites are accessible for CaM (only one binding event is shown here for clarity). (B) Same as in (A) for TRPV5. The two-lobed CaM binds into the central cytoplasmic cavity of TRPV5. While four potential binding sites are available in TRPV5, only a single CaM molecule can bind due to steric hindrance. Unlike TRPC4, in which the C-terminal helices block the access to the cytoplasmic cavity, CaM can enter into the internal cavity of TRPV5 from the cytoplasm. (C) Same as in (A) for TRPV6. Similar to TRPV5, only a single CaM molecule binds to a region within the cytoplasmic cavity of TRPV6, indicating that this binding mode is conserved among TRPV channels.

## Model for TRPC4 modulation

In this study we determined the structure of TRPC4 in complex with the pyridazinone-based inhibitors GFB-8438, GFB-9289, and GFB-8749 as well as with its endogenous regulator CaM. Analysis of these structures allows us to propose a model describing the molecular mechanism of modulation and regulation of TRPC4 activity (*Figure 7A*). In our model, the channel switches between its closed and open conformation as was proposed for many other channels. Upon binding of an activator, the channel opens transiently and allows the passage of $Ca^{2+}$. Binding of inhibitors locks the channel in its closed conformation and possibly results in a dissociation of the activator. In our case, all of the three inhibitors bind to the same position, namely the VSL domain which is connected to the gate by the TRP helix. Thus, subtle conformational changes in this sensitive regulatory domain appear to be sufficient to transfer the signal from the periphery to the center of the channel to modulate its activity. High concentrations of the activator can reverse this effect (*Figure 1*).

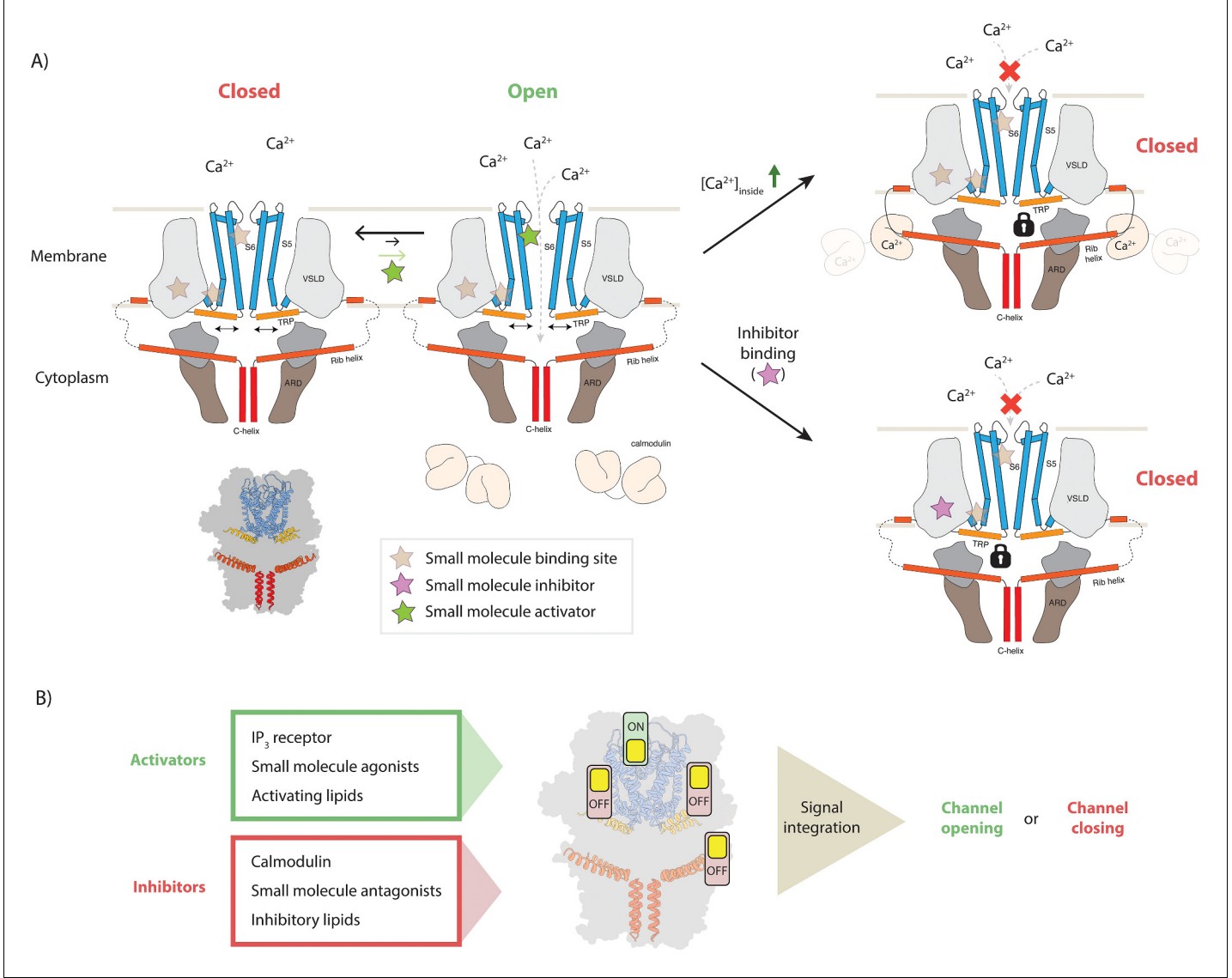

**Figure 7.** Model for TRPC4 modulation. (**A**) Canonical TRP channels can transiently open to allow the passage of $Ca^{2+}$ ions into the interior compartment (left panel). Several mechanisms modulate the activity of the channel: binding of small molecule activators to one of the ligand binding pockets favors the opening event and thereby increases the overall channel activity. In the gating process, the TRP helix (orange) plays a central role as it has a direct connection to the pore-forming helices (blue), constituting the ion-conducting pore. Binding of small molecule inhibitors and the inhibitory protein CaM can restrict the mobility of the TRP helix, thus locking the channel in the closed state (bottom and top panels on the right, respectively). In the latter case, high intracellular $Ca^{2+}$ concentrations cause the $Ca^{2+}$-sensing protein CaM to bind to the CIRB region of the protruding rib-helix (red). This binding event stabilizes a previously disordered region that directly connects to the TRP helix. (**B**) Individual or simultaneous binding of activators and/or inhibitors modulate the channel gating. Interestingly, modulation sites, i.e. ligand pockets or structural features to which certain compounds or regulatory proteins bind, can accommodate both activators and inhibitors. Thus, these regions can be considered as activity switches. Binding of activators results in an 'ON' position, whereas inhibitor binding causes an 'OFF' state. In the case that multiple modulators bind simultaneously, all signals are integrated to determine whether the channel opens or remains closed.

CaM does not bind to the VSL domain, which resides in the membrane and is therefore not directly accessible. However, it stabilizes other parts of the protein that are connected to the VSL domain. In particular, it binds to the tip of the rib helix, which results in the stabilization of the loop and the helix that connects it to the VSL domain. Thus, the binding of CaM to the rib helix has the same consequence as the binding of an inhibitor to the VSL domain, locking the channel in its closed conformation. Interestingly, the rib helix has also been shown to be the binding site for the IP$_3$

receptor which acts as an activator of TRPC4 (*Tang et al., 2001*). Although we do not yet know the structural details of this interaction, it is likely that inhibition by CaM and activation by IP$_3$ receptor require the use the same binding site, while resulting in opposing effects. Activation by DAG (*Storch et al., 2017*) likely happens by binding of the secondary messenger to the previously identified lipid binding site (*Vinayagam et al., 2018*) close to the channel pore. Thus, DAG activation would act directly on the pore region to open the channel. A structure of TRPC4 in complex with DAG will hopefully elucidate the exact mechanism in the near future.

Thus, TRPC4 contains several molecular switch regions that can be modulated by the binding of small molecules or regulatory proteins (*Figure 7B*). Consequently, the signals of different switches are integrated and together determine the final state and the degree of activation of the channel. Our model not only explains how TRPC4 activity is regulated by CaM in the cellular context, but also opens new possibilities for knowledge-driven pharmacological manipulation of this therapeutic target.

# Materials and methods

## Key resources table

| Reagent type (species) or resource | Designation | Source or reference | Identifiers | Additional information |
|---|---|---|---|---|
| Cell line (HEK293 GnTI-) | HEK293 GnTI- | ATCC | RRID:CVCL_A785 | CRL-3022 |
| Cell line (HEK293T) | HEK293T | ATCC | RRID:CVCL_LF41 | |
| Cell line (Sf9) | Sf9 | Oxford Expression Technologies Ltd (UK) | RRID:CVCL_0549 | Cat.No.600100 |
| Gene (*Danio rerio*) | TRPC4$_{DR}$ | GenScript NCBI Reference sequence: NM_001289881 | | |
| Recombinant DNA reagent | pCDNA3.1+TRPC4$_{ZF}$ | *Vinayagam et al., 2018* PMID:29717981 | | |
| Recombinant DNA reagent | pEG BacMam | Eric Gouaux Lab PMID:25299155 | | |
| Recombinant DNA reagent | pEG BacMam +TRPC4 $_{ZF}$ (See methods section for details) | *Vinayagam et al., 2018* PMID:29717981 | | |
| Recombinant DNA reagent | pGEMHE 22 | Promega | P2151 | pGEMHE 22 is a derivative of pGEM3z |
| Chemical compound, drug | (-)-Englerin A | Carl Roth | Cat.No.6492.1 | |
| Software, algorithm | SPHIRE software package | *Moriya et al., 2017* PMID:28570515 | | |
| Software, algorithm | crYOLO | *Wagner et al., 2019* PMID:31240256 | | |
| Software, algorithm | Origin 2020 pro | OriginLab Corporation | | |
| Software, algorithm | TranSPHIRE | *Stabrin et al., 2020* doi:https://doi.org/10.1101/2020.06.16.155275 | | |
| Software, algorithm | Chimera | *Pettersen et al., 2004* PMID:15264254 | | |

## Protein purification and expression

Zebrafish TRPC4$_{DR}$ was prepared as described previously (*Vinayagam et al., 2018*). In brief, residues 2–915 of *Danio rerio* TRPC4 were cloned into the pEG BacMam vector (*Goehring et al., 2014*), with a C-terminal HRV-3C cleavage site followed by EGFP and a twin StrepII-tag. An 8x His-tag with a TEV cleavage site was positioned at the N-terminus. Baculovirus was produced as described previously (*Goehring et al., 2014*). P2 baculovirus produced in Sf9 cells was added to HEK293 GnTI$^-$ cells (mycoplasma test negative, ATCC #CRL-3022) and grown in suspension in FreeStyle medium (GIBCO-Life Technologies) supplemented with 2% FBS at 37°C and 8% CO$_2$. After 8 hr of

transduction 5 mM sodium butyrate was added to enhance protein expression and allowed the cells to grow for an additional 40 hr at 30°C.

48 hr post transduction, cells were harvested by centrifugation at 1,500 $g$ for 10 mins and washed in phosphate-buffered saline (PBS) pH 7.4. The cell pellet was resuspended and cells were lysed in an ice-cooled microfluidizer in buffer A (PBS buffer pH 7.4, 1 mM Tris(2-carboxyethyl) phosphine (TCEP), 10% glycerol) in the presence of protease inhibitors (0.2 mM AEBSF, 10 μM leupeptin). 50 ml buffer A was used per pellet obtained from 800 ml of HEK293 cell culture. The lysate was centrifuged at 5,000 $g$ for 5 min to remove cell debris, followed by a 15,000 g centrifugation for 10 mins to remove sub-cellular organelles. The membranes were collected by ultracentrifugation using a Beckman Coulter Type 70 Ti rotor at 40,000 rpm. The membranes were then mechanically homogenized in buffer B (100 mM Tris-HCl pH 8, 150 mM NaCl, 1 mM TCEP, 10% glycerol) containing protease inhibitors, flash-frozen and stored at −80°C until further purification.

## Purification of TRPC4 in DDM followed by amphipol exchange

Membranes were solubilized for 2 hr in buffer B supplemented with 1% dodecyl maltoside (Anatrace #D310). Insoluble material was removed by ultracentrifugation for 1 hr in a Beckman Coulter Type 70 Ti rotor at 40,000 rpm. The soluble membrane fraction was diluted 2-fold with buffer B and applied to a column packed with Strep-Tactin beads (IBA Lifesciences) by gravity flow (6–10 s/drop) at 4°C. Next, the resin was washed with ten column volumes of buffer B supplemented with 0.04% DDM solution containing protease inhibitors. Bound protein was eluted seven times with 0.5 column volumes of buffer A with 3 mM d-Desthiobiotin (Sigma-Aldrich), 0.026% DDM and 0.1 mM AEBSF protease inhibitor. The C-terminal EGFP tag was removed by incubating the eluted fractions with HRV-3C protease overnight. The next day, the detergent was replaced with amphipol A8-35 (Anatrace) 4:1 (w/w) to the cleaved protein and incubating for 6 hr at 4°C. Detergent removal was performed by adding Biobeads SM2 (BioRad) pre-equilibrated in PBS to the protein solution at 10 mg/ml final concentration for 1 hr, then replaced with fresh Biobeads at 10 mg/ml for overnight incubation at 4°C. Biobeads were removed using a Poly-Prep column (BioRad) and the solution was centrifuged at 20,000 g for 10 min to remove any precipitate. The protein was concentrated with a 100 MWCO Amicon centrifugal filter unit (Millipore) and purified by size exclusion chromatography using a Superose 6 Increase 10/300 GL column (GE Healthcare) equilibrated in buffer C (PBS pH 7.4, 1 mM TCEP). The peak corresponding to tetrameric TRPC4$_{DR}$ in amphipols was collected and analyzed initially with negative stain EM and then by cryo-EM.

## TRPC4 pulldown assay using CaM sepharose beads

The assay was performed with manufacturer instructions using CaM as bait to bind TRPC4. Briefly, 1 ml of CaM sepharose beads were loaded into the Biorad Ployprep column and washed with 10 ml of binding buffer containing 20 mM Tris-HCl (pH 7.5), 150 mM NaCl, 2 mM CaCl$_2$. TRPC4 prepared in LMNG (described below) was loaded onto the column by gravity flow (10 s/drop) at 4°C. After loading, the column was washed with 10 ml of binding buffer. Finally, TRPC4 was eluted with 5 ml of elution buffer containing 20 mM Tris-HCl (pH 7.5), 150 mM NaCl, 2 mM EGTA (*Figure 5—figure supplement 1*).

## Purification of CaM

Mouse CaM was subcloned into a pET19 vector and expressed in BL21-CodonPlus (DE3) -RIPL cells. Cells were grown in LB broth with 125 μg/ml ampicillin at 37°C until an OD600 of 0.4 was reached. Subsequently, CaM expression was induced with 1 mM IPTG and grown overnight at 20°C. Cells were harvested by centrifugation and resuspended in 50 ml (per liter of culture) of lysis buffer containing 20 mM Tris-HCl (pH 8.0), 150 mM NaCl and 5 mM imidazole. The cells were lysed in an ice-cooled microfluidizer. The soluble fraction obtained after centrifugation was loaded onto an 8 ml Talon resin column pre-equilibrated with lysis buffer. The resin was washed with 100 ml of lysis buffer containing 20 mM Tris-HCl (pH 8.0), 150 mM NaCl and 20 mM imidazole before eluting in 5 × 5 ml fractions using 25 ml of lysis buffer supplemented with 20 mM Tris-HCl (pH 8.0), 150 mM NaCl and 250 mM imidazole. CaM was further purified by size exclusion chromatography using a Superose 12 10/300 gel filtration column and stored at −80°C in a storage buffer consisting of 20 mM Tris-HCl (pH 8.0), 150 mM NaCl, 10% glycerol.

To check the interaction of N- and C-lobe of CaM with TRPC4, residues 1–80 and 81–149 of CaM (forming the N-lobe and C-lobe respectively) were individually cloned into a pMAL vector as an MBP-fusion construct along with N-terminal His-tag. The clones were expressed in BL21-CodonPlus (DE3)-RIL cells. The cells were grown in LB broth with 125 μg/ml ampicillin and 34 μg/ml chloramphenicol at 37°C until an OD600 of 0.6 was reached. Then, CaM expression was induced with 0.2 mM IPTG and grown overnight at 19°C. Cells were harvested by centrifugation and resuspended in 100 ml (per 5 liter of culture) of lysis buffer containing 50 mM Tris-HCl (pH 8.0), 150 mM NaCl and 10% glycerol with the addition of Roche protease inhibitor cocktail, 2 mM β-mercaptoethanol and 0.1% NP40. The soluble fraction obtained after centrifugation was loaded onto $2 \times 5$ ml $Ni^{2+}$ HisTrap HP columns pre-equilibrated with lysis buffer. Next the column was washed with wash buffer containing 50 mM Tris-HCl (pH 8.0), 500 mM NaCl, 10% glycerol, 10 mM imidazole 2 mM β-mercaptoethanol. Finally, the protein was eluted in a gradient fashion using lysis buffer supplemented with 500 mM imidazole. The peak eluted fractions were further concentrated and injected into a Superdex 75 16/60 gel filtration column. The protein purified after gel filtration was used for a pulldown experiment to screen for TRPC4 interaction.

## TRPC4 pull-down assay using CaM N- and C-lobe MBP-fusion constructs

The assay was done using amylose resin using the N- and C-lobe of CaM as a bait to pulldown TRPC4. 100 μl of amylose resin was pipetted into a mini spin column. The resin was washed with 500 μl of equilibration buffer containing 50 mM Tris-HCl (pH 8.0), 150 mM NaCl, 100 μM $CaCl_2$, 0.003% LMNG. Affinity purified TRPC4 at concentration of 0.2 mg/ml was mixed in a separate tube with 1.5 mg/ml N- and C-lobe CaM fused to MBP, and incubated for 30 min at 4°C. 200 μl of the TRPC4-CaM N- and C-lobe complexes were applied to the equilibrated resin in different tubes. The columns were washed with 500 μl equilibration buffer and eluted with 100 μl of equilibration buffer supplemented with 10 mM maltose. 20 μl aliquots were taken from each step and analyzed using SDS-PAGE for TRPC4 interaction with the separate CaM N- and C-lobe (*Figure 5—figure supplement 6*).

## Preparation of the TRPC4-CaM complex

TRPC4 membranes were solubilized for 2 hr in buffer B supplemented with 1% LMNG (Anatrace #NG310). Then a protocol similar to that used for DDM purification was followed, except that DDM in buffer B was replaced by LMNG with the addition of 10 μM CaM and 10 mM calcium chloride. The LMNG detergent concentration was maintained at five times the CMC for washing buffer and three times CMC for elution. The C-terminal EGFP tag was removed by incubating the eluted fractions with HRV-3C protease overnight. The complex was further purified by size exclusion chromatography using a Superose 6 Increase 10/300 GL column (GE Healthcare) equilibrated in buffer containing 20 mM Tris-HCl (pH 8.0), 150 mM NaCl, 1 mM TCEP, 10 mM calcium chloride and 5% glycerol. Complex formation was assessed by running SDS-PAGE of the peak fraction known to contain TRPC4 (*Figure 5—figure supplement 1*). The gel analysis indicated sub-saturation of the complex. Hence, 10 μM CaM was added to saturate the complex before concentrating it to 0.3 mg/ml for plunging. The preparation of TRPC4 -apo in LMNG was similar to the TRPC4-CaM complex except that CaM and $CaCl_2$ were not added.

## Cryo-EM grid preparation and screening

The sample quality and integrity were evaluated by negative stain electron microscopy prior to cryo-EM grid preparation and image acquisition as described earlier (*Vinayagam et al., 2018*). Typically, 4 μl of TRPC4$_{DR}$ at a sample concentration of 0.02 mg/ml was applied onto a freshly glow-discharged copper grid with an additional thin carbon layer. After incubation for 45 s, the sample was blotted with Whatman no. 4 filter paper and stained with 0.75% uranyl formate. The images were recorded manually with a JEOL JEM-1400 TEM operated at an acceleration voltage of 120 kV, and a 4k F416 CMOS detector (TVIPS). For cryo-EM the ligands dissolved in DMSO were added to a final concentration of 100 μM (final DMSO concentration 1%) to TRPC4 exchanged in amphipols and incubated for 30 min before plunging using a Vitrobot cryo-plunger (FEI Thermo Fisher) operated at 4°C and 100% humidity. Details of the plunging conditions are summarized in *Table 1*.

**Table 1.** Plunging and imaging conditions used for cryo-EM analysis of TRPC4 bound with ligands.

**1.1 Plunging conditions**

| Sample | Grid type | Volume | Concentration | Blotting time | Blotting force |
|---|---|---|---|---|---|
| TRPC4-8438 | C-Flat 2/1 | 3 µl | 0.3 mg/ml | 3 s | −10 |
| TRPC4-9289 | C-Flat 1.2/1.3 | 3 µl | 0.35 mg/ml | 3 s | 0 |
| TRPC4-8749 | C-Flat 1.2/1.3 | 3 ul | 0.35 mg/ml | 3s | 0 |
| TRPC4-cam | QF 2/1 | 3 µl | 0.3 mg/ml | 3 s | 0 |
| TRPC4-apo(LMNG) | C-Flat 1.2/1.3 | 3 µl | 0.4 mg/ml | 3 s | −3 |

**1.2 Imaging Conditions**

| Microscopy | TRPC4-apo | TRPC4-CaM | GFB-9289 | GFB-8438 | GFB-8749 |
|---|---|---|---|---|---|
| Microscope | Titan Krios (X-FEG, Cs-corrected) | | Titan Krios (X-FEG, Cs 2.7 mm) | | |
| Voltage [kV] | 300 | | 300 | | |
| Defocus range [µm] | 0.65 to 3.02 | 0.38 to 3.48 | 0.68 to 3.64 | 0.35 to 3.52 | 0.86 to 3.82 |
| Camera | K2 counting | K2 counting | K3 Super res. | K3 Super res. | K3 Super res. |
| Pixel size [Å] | 0.85 | 0.85 | 0.455 /0.91[a] | 0.455/0.91[a.] | 0.455 /0.91[a] |
| Total electron dose [e/Å$^2$] | 88.7 | 88.2 | 65.45 | 66.58 | 72 |
| Exposure time [s] | 10 | 10 | 3 | 3 | 3 |
| Frames per movie | 50 | 80 | 60 | 60 | 60 |
| Number of images | 2755 | 6937 | 2369 | 4444 | 1260 |
|  | (3079) | (7972) | (2970) | (4676) | (1290) |

## Cryo-EM data acquisition and image processing

Data sets were collected using EPU software on Titan Krios microscopes (FEI Thermo Fisher) operated at 300 kV and equipped with an X-FEG. For the dataset of the GFB-9289-bound TRPC4 the aberration-free image shift (AFIS) feature of EPU was used to speed up the data-collection process. Equally dosed frames were collected using a K2 Summit (Gatan) or K3 (Gatan) direct electron detectors in super-resolution mode in combination with a GIF quantum-energy filter set to a filter width of 20 eV. The details of all four data sets including pixel size, electron dose, exposure time, number of frames and defocus range are summarized in *Table 1*. Data collection was monitored live using TranSPHIRE (*Stabrin et al., 2020*), allowing for direct adjustments of data acquisition settings when necessary, i.e. defocus range or astigmatism. The total number of images collected is summarized in *Table 1*. Preprocessing included drift correction with MotionCor2 (*Zheng et al., 2017b*), creating aligned full-dose and dose-weighted micrographs. The super-resolution images were binned twice after motion correction to speed up further processing steps. CTF estimation was also performed within TranSPHIRE using CTFFIND 4.1.10 (*Rohou and Grigorieff, 2015*) on non-dose-weighted aligned micrographs. Unaligned frame averages were manually inspected and removed based on ice and image quality, resulting in a removal of 5–20% of the data sets (see *Table 1* for details). Following processing steps were performed using motion-corrected dose-weighted sums in the SPHIRE software package unless otherwise indicated (*Moriya et al., 2017*).

Single particles were picked automatically with crYOLO using the general model (*Wagner et al., 2019*). The particles were then windowed to a final box size of 288 × 288 pixels. Reference-free 2-D classification and cleaning of the data set was performed with the iterative stable alignment and clustering approach ISAC (*Yang et al., 2012*) in SPHIRE. ISAC was performed at a pixel size of 3.52 Å/pixel for apo-TRPC4 and TRPC4 bound to CaM. For inhibitors bound TRPC4 ISAC run was performed with either 3.52 Å/pixel or 3.8 Å/pixel. The 'Beautify' tool of SPHIRE was then applied to obtain refined and sharpened 2-D class averages at the original pixel size, showing high-resolution features. A subset of particles producing 2-D class averages and reconstructions with high-resolution features were then selected for further structure refinement. The previously reported apo structure was used as reference for 3D refinement in MERIDIEN with imposed C4 symmetry (*Moriya et al.,*

*2017*). Further polishing and CTF refinement were carried out in RELION 3.0.4 (*Zivanov et al., 2018*).

In case of GFB-9289 bound to TRPC4 bound structure, the refinement did not improve above 4.1 Å, as the dataset collected with AFIS suffered from strongbeam tilt which was estimated and corrected in RELION before 3D classification. For both the ligands, a 3D classification was performed with C4 symmetry to classify the subpopulation. The classes having high-resolution features bound with ligands were selected and further polished and CTF-refined in RELION.

For TRPC4 bound with CaM, 3D classification using Sort3d in SPHIRE was performed to identify subpopulations with different stoichiometries. To further improve the resolution of the CaM region, we used symmetry expansion by quadrupling the 227,693 particles to reflect the C4 symmetry of the tetramer. Thus, the resulting 910,772 particles were used for Sort3d with a focused mask comprising the four CaM regions without imposing symmetry. Ten different classes obtained with Sort3d showed different stoichiometries (TRPC4 monomer:CaM) as shown in *Figure 5—figure supplement 2*. Four classes showing well resolved helices for CaM were selected and oriented in the same direction in order to boost the density at single CaM site (*Figure 5—figure supplement 2*). This rotation was achieved by applying a rotation of (±90˚, 180˚, 270˚) to the projection parameters of the classes using a customized script. After rotation, duplicates were removed, reducing the number of particles to 160,829. These particles were further polished and CTF-refined in RELION. The polished particles were finally refined in MERIDIEN (SPHIRE) with C1 symmetry using a mask encompassing TRPC4 with a single CaM.

## Local resolution estimation and filtering

The final half-maps were combined using a tight mask with the application of B-factors automatically determined by the PostRefiner tool in SPHIRE and filtered to the estimated resolution. The final estimated resolution by the 'gold standard' FSC = 0.143 criterion between the two masked half-maps is given in *Table 2*. The local resolution was calculated using sp_locres in SPHIRE. In case of TRPC4-CaM, the final densities were filtered according to local resolution using the local de-noising filter LAFTER (*Ramlaul et al., 2019*) for the purpose of model building.

## Model building, refinement and validation

The previously reported model of TRPC4 (*Vinayagam et al., 2018*) was initially docked into the density and fitted into the map as rigid body using UCSF Chimera. The model was further adjusted to fit in the density using Coot (*Emsley et al., 2010*) with an iterative process of real space refinement in Phenix (*Adams et al., 2010*) and model adjustment in Coot until convergence as evaluated by model-to-map fit with valid geometrical parameters. The high resolution obtained with the GFB-9289 and apo structure enabled accurate modelling of the structure especially in the region encompassing residues 727–731 that connects the rib helix to the C-terminal helix (*Figure 5—figure supplement 3*). The presence of connecting density at this region shows the swapping of helices in this region. In our previous model, the density for the corresponding area was less resolved and the C-terminal helix was modelled without the domain swapping of the C-terminal helix. For the inhibitor molecules, cif files were generated using eLBOW tool in Phenix and used as geometrical restraints in Coot and Phenix during modelling and refinement respectively.

In the TRPC4-CaM complex, both the N- and C-lobes of CaM-bound with myosin light chain kinase (PDB ID: 2LV6) were separately used for rigid body fitting into the CaM density using Chimera. Both the lobes fit to the density using rigid body fitting which agrees with the results of the pull-down experiment (*Figure 5—figure supplement 6*) and corroborates a previous study (*Zhu, 2005*). The N- and C-lobes were then flexibly fitted into the density with the Cryo_fit tool in Phenix, which employs MD simulations. The C-terminal lobe fit into density better than the N-terminal lobe, hence the C-terminal lobe was used for modelling into the density. Using this CaM model as an initial guide, the CaM was further manually adjusted to fit inside the density using Coot. Several rounds of iterative model building and refinement were performed using Coot and Phenix respectively until a good fit with a valid geometry was obtained (*Table 2*).

The densities corresponding to annular lipids were modelled as phosphatidic acid lipid (PDB ligand ID LPP) in the structures of GFB9289-bound TRPC4 and apo TRPC4. In case of CaM-bound

**Table 2.** Refinement and model validation statistics.

**Refinement statistics**

| | TRPC4-apo | TRPC4-CaM | GFB- 9289 | GFB-8438 | GFB-8749 |
|---|---|---|---|---|---|
| Number of particles | | | | | |
| used in refinement | 126873 | 160829 | 65811 | 42524 | 44989 |
| Final resolution [Å] | 2.8 | 3.6 | 3.2 | 3.6 | 3.8 |
| Map sharpening factor [Å²] | -57.97 | -72.37 | -100 | -61.35 | -120 |
| Electron dose particles final refinement [e⁻/Å²] | Polished particles | Polished particles | Polished particles | Polished particles | 72 |
| Model geometry and validation statistics | | | | | |
| Atomic model composition | | | | | |
| Non-hydrogen atoms | 22,124 | 21,650 | 21,152 | 21,080 | 21,056 |
| Refinement (Phenix) | | | | | |
| RMSD bond | 0.008 | 0.011 | 0.008 | 0.007 | 0.011 |
| RMSD angle | 0.738 | 0.983 | 0.645 | 0.771 | 0.736 |
| Model-to-map fit, CC mask | 0.84 | 0.86 | 0.85 | 0.86 | 0.83 |
| Validation Ramachandran plot (%) | | | | | |
| Outliers | 0.0 | 0.04 | 0.0 | 0.0 | 0.0 |
| Allowed | 7.44 | 9.91 | 7.15 | 5.72 | 7.9 |
| favored | 92.56 | 90.05 | 92.85 | 94.28 | 91.94 |
| Rotamer outliers (%) | 0.51 | 0.09 | 8.99 | 0.35 | 0.18 |
| Molprobity score | 1.82 | 2.29 | 2.38 | 1.84 | 1.99 |
| EMRinger score | 3.04 | 1.61 | 2.67 | 2.28 | 2.75 |

TRPC4 and other inhibitor-bound TRPC4, a shorter lipid tail (PDB ligand ID 44E) was modelled due to limited resolution.

Finally, validation statistics computed by Phenix using MolProbity (*Chen et al., 2010*) were used to validate the overall geometry of the model, the model-to-map correlation value to assess the fitness of the model to its density, and an EMRinger score (*Barad et al., 2015*) to validate side chain geometry.

Figures were prepared in Chimera (*Pettersen et al., 2004*). Multiple sequence alignment was done using Clustal Omega (*Sievers et al., 2011*). Figures of the sequence alignment were made in Jalview (*Waterhouse et al., 2009*). The radius of the TRPC4 pore was determined using HOLE (*Smart et al., 1996*).

## Synthesis of GFB-9289

### 4-chloro-5-(4-cyclohexyl-3-oxopiperazin-1-yl)-2,3-dihydropyridazin-3-one (GFB-9289)

To a solution of 1-cyclohexylpiperazin-2-one (150 mg, 0.8 mmol, one equivalent) in DMF (5 mL) was added 4,5-dichloro-2,3-dihydropyridazin-3-one (410 mg, 2.5 mmol, 3.0 equivalent) and DIEA (442 mg, 3.4 mmol, 4.0 equivalent) at ambient temperature under air atmosphere. The resulting mixture was stirred for 5 hr at 100°C. Then the reaction mixture was cooled and purified by reverse phase flash with the following conditions (Column: C18 OBD Column, 5 um, 19 × 330 mm; Mobile Phase A: Water (5 mmol/L $NH_4HCO_3$), Mobile Phase B: ACN; Flow rate: 45 mL/min; Gradient: 30% B to 60% B in 40 min; 254 nm; Rt: 15 min) to afford crude product (80 mg), which was further purified by Chiral-Prep-HPLC with the following conditions: Column: CHIRALPAK IG-3, Column size: 0.46 × 5 cm;3 um; Mobile phase: Hex(0.1%DEA):EtOH = 80:20; Pressure: MPA; Flow: 1.0 ml/min; Instrument: LC-08; Detector: 254 nm; Temperature: 25°C. 4-chloro-5-(4-cyclohexyl-3-oxopiperazin-1-yl)−2,3-dihydropyridazin-3-one (26.5 mg, 10.4%) was obtained at 1.436 min as a white solid (26.5 mg). [1]H NMR

(400 MHz, DMSO-$d_6$) chemical shifts δ 12.91 (s, 1H), 7.86 (s, 1H), 4.23 (t, $J$ = 12.1 Hz, 1H), 4.09 (s, 2H), 3.68 (t, $J$ = 5.2 Hz, 2H), 3.38 (t, $J$ = 5.3 Hz, 2H), 1.77 (d, $J$ = 12.8 Hz, 2H), 1.61 (d, $J$ = 15.6 Hz, 2H), 1.58–1.40 (m, 3H), 1.31 (q, $J$ = 13.1 Hz, 2H), 1.11 (t, $J$ = 13.1 Hz, 1H). LRMS (ESI) $m/z$: [M+H]$^+$ calculated for $C_{14}H_{20}ClN_4O_2$ 311.13; found 311.15. Purity 96%.

## Electrophysiology assay

TRPC4-GFP DNA fragments were inserted into pGEMHE 22. The complementary RNA (cRNA) was synthesized by in vitro transcription using the AmpliCap-MaxT7 High Yield Message Maker Kit (Epicentre Biotechnologies) and stored in nuclease-free water at −20°C. Stage V and VI oocytes were surgically removed from female *Xenopus laevis* by immersion in water containing 1 g/L Tricain and isolated from theca and follicle layers by digestion with 0.14 mg ml$^{-1}$ collagenase I. Oocytes were injected with 8 ng cRNA and were incubated at 16°C for 3 days in ND96 solution (96 mM NaCl, 2 mM KCl, 1 mM CaCl$_2$, 1 mM MgCl$_2$, 10 mM HEPES, pH7.4). Two-electrodes voltage clamp measurements with Xenopus oocytes were performed at room temperature (20–23°C) in modified standard Ringer's solution (110 mM NaCl, 5 mM KCl, 2 mM BaCl$_2$ (to avoid Ca$^{2+}$-activated current of endogenous chloride channels), 1 mM MgCl$_2$, 10 mM HEPES, pH 7.4) with a TURBO TEC-03X amplifier (npi electronic GmbH, Tamm, Germany). Electrode capillaries (Φ=1.5 mm, Hilgenberg) were filled with 3 M KCl, with tip resistances of 0.4–1 MΩ. USB-6221 DAQ device (National Instruments) and WinWCP (v5.5.3, Strathclyde University, UK) are used for data acquisition.

## Acknowledgements

We are thankful to Nina Ludwigs, Marion Hülseweh and Nathalie Bleimling for technical assistance. We also thank Ingrid Vetter and Sabrina Pospich for helpful discussion with model building and analysis and Amrita Rai for assay development. We thank Dr. Shiqiang Gao for subcloning TRPC4-GFP into pGEMHE for oocyte expression. We are grateful to Matthew Daniels for assistance with compound comparison and selection. This work was supported by funds from the Max Planck Society (to SR). JY-S was supported by TR240 from the DFG (to GN).

## Additional information

### Competing interests

Maolin Yu, Mark W Ledeboer, Goran Malojcic: The author is or was a shareholder of Goldfinch Bio. The other authors declare that no competing interests exist.

### Funding

| Funder | Grant reference number | Author |
|---|---|---|
| Max-Planck-Gesellschaft | | Stefan Raunser |
| Deutsche Forschungsgemeinschaft | TR240 | Georg Nagel |

The funders had no role in study design, data collection and interpretation, or the decision to submit the work for publication.

### Author contributions

Deivanayagabarathy Vinayagam, Data curation, Formal analysis, Investigation, Visualization, Writing - original draft; Dennis Quentin, Visualization, Writing - original draft; Jing Yu-Strzelczyk, Data curation, Formal analysis, Visualization; Oleg Sitsel, Data curation, Investigation, Visualization, Writing - original draft; Felipe Merino, Validation, Investigation; Markus Stabrin, Software, Formal analysis; Oliver Hofnagel, Maolin Yu, Data curation; Mark W Ledeboer, Project administration, Writing - review and editing; Georg Nagel, Data curation, Formal analysis, Funding acquisition; Goran Malojcic, Conceptualization, Formal analysis, Project administration, Writing - review and editing; Stefan Raunser, Conceptualization, Supervision, Funding acquisition, Validation, Writing - original draft, Project administration, Writing - review and editing

## Author ORCIDs
Deivanayagabarathy Vinayagam (iD) https://orcid.org/0000-0001-7862-9499
Dennis Quentin (iD) https://orcid.org/0000-0003-3825-7066
Oleg Sitsel (iD) https://orcid.org/0000-0002-4496-7489
Felipe Merino (iD) http://orcid.org/0000-0003-4166-8747
Markus Stabrin (iD) http://orcid.org/0000-0003-0191-6419
Stefan Raunser (iD) https://orcid.org/0000-0001-9373-3016

## Decision letter and Author response
Decision letter https://doi.org/10.7554/eLife.60603.sa1
Author response https://doi.org/10.7554/eLife.60603.sa2

---

# Additional files

## Supplementary files
- Supplementary file 1. Supplementary method for the synthesis of GFB-8749.

- Transparent reporting form

## Data availability
The atomic coordinates and cryo-EM maps for TRPC4DR in complex with inhibitors, calmodulin and for TRPC4DR in LMNG are available at the Protein Data Bank (PDB) and Electron Microscopy Data Bank (EMDB) databases, under the accession numbers PBD 7B0S and EMD-11970 (TRPC4-GFB8438), PBD 7B16 and EMD-11979 (TRPC4-GFB9289); PBD 7B05 and EMD-11957 (TRPC4-GFB8749); PBD 7B1G and EMD-11985 (TRPC4-Calmodulin) and PBD 7B0J and EMD-11968 (TRPC4-apo in LMNG).

The following datasets were generated:

| Author(s) | Year | Dataset title | Dataset URL | Database and Identifier |
|---|---|---|---|---|
| Vinayagam D, Quentin D, Sitsel O, Merino F, Stabrin M, Hofnagel O, Lede-boer MW, Malojcic G, Raunser S | 2020 | TRPC4-GFB8438 | https://www.rcsb.org/structure/7B0S | RCSB Protein Data Bank, 7B0S |
| Vinayagam D, Quentin D, Sitsel O, Merino F, Stabrin M, Hofnagel O, Lede-boer MW, Malojcic G, Raunser S | 2020 | TRPC4-GFB9289 | https://www.rcsb.org/structure/7B16 | RCSB Protein Data Bank, 7B16 |
| Vinayagam D, Quentin D, Sitsel O, Merino F, Stabrin M, Hofnagel O, Lede-boer MW, Malojcic G, Raunser S | 2020 | TRPC4-GFB8749 | https://www.rcsb.org/structure/7B05 | RCSB Protein Data Bank, 7B05 |
| Vinayagam D, Quentin D, Sitsel O, Merino F, Stabrin M, Hofnagel O, Lede-boer MW, Malojcic G, Raunser S | 2020 | TRPC4-Calmodulin | https://www.rcsb.org/structure/7B1G | RCSB Protein Data Bank, 7B1G |
| Vinayagam D, Quentin D, Sitsel O, Merino F, Stabrin M, Hofnagel O, Lede-boer MW, Malojcic G, Raunser S | 2020 | TRPC4-apo in LMNG | https://www.rcsb.org/structure/7B0J | RCSB Protein Data Bank, 7B0J |

| Vinayagam D, Quentin D, Sitsel O, Merino F, Stabrin M, Hofnagel O, Ledeboer MW, Malojcic G, Raunser S | 2020 | TRPC4-GFB8438 | https://www.ebi.ac.uk/pdbe/entry/emdb/EMD-11970 | Electron Microscopy Data Bank, 11970 |
|---|---|---|---|---|
| Vinayagam D, Quentin D, Sitsel O, Merino F, Stabrin M, Hofnagel O, Ledeboer MW, Malojcic G, Raunser S | 2020 | TRPC4-GFB9289 | https://www.ebi.ac.uk/pdbe/entry/emdb/EMD-11979 | Electron Microscopy Data Bank, 11979 |
| Vinayagam D, Quentin D, Sitsel O, Merino F, Stabrin M, Hofnagel O, Ledeboer MW, Malojcic G, Raunser S | 2020 | TRPC4-GFB8749 | https://www.ebi.ac.uk/pdbe/entry/emdb/EMD-11957 | Electron Microscopy Data Bank, 11957 |
| Vinayagam D, Quentin D, Sitsel O, Merino F, Stabrin M, Hofnagel O, Ledeboer MW, Malojcic G, Raunser S | 2020 | TRPC4-Calmodulin | https://www.ebi.ac.uk/pdbe/entry/emdb/EMD-11985 | Electron Microscopy Data Bank, 11985 |
| Vinayagam D, Quentin D, Sitsel O, Merino F, Stabrin M, Hofnagel O, Ledeboer MW, Malojcic G, Raunser S | 2020 | TRPC4-apo in LMNG | https://www.ebi.ac.uk/pdbe/entry/emdb/EMD-11968 | Electron Microscopy Data Bank, 11968 |

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
