## [Decision Letter]

**Acceptance summary:**

The present manuscript describes structures of TRPC4 channels in the presence of new inhibitory molecules. The data is of sufficient quality to unambiguously show specific interactions between the channel and the modulatory compounds. The authors also show a structure of the channel interacting with calmodulin and describe this interaction as inhibitory. These new structures are an important step in understanding the mechanisms of pharmacological regulation of TRP channels and TRPC4 in particular.

**Decision letter after peer review:**

Thank you for submitting your article "Structural basis of TRPC4 regulation by calmodulin and pharmacological agents" for consideration by *eLife*. Your article has been reviewed by three peer reviewers, including Leon D Islas as the Reviewing Editor and (Reviewer #1), and the evaluation has been overseen by Olga Boudker as the Senior Editor. The following individual involved in review of your submission has agreed to reveal their identity: Vera Y Moiseenkova-Bell (Reviewer #2).

The reviewers have discussed the reviews with one another and the Reviewing Editor has drafted this decision to help you prepare a revised submission.

Summary:

In this manuscript, Vinayagam et al. reports several cryo-EM derived structures of the closed TRPC4 channel in complex with small inhibitor GFB-9289, activator GFB-8438 and calmodulin. The authors analyzed these structures to propose specific amino acid residues forming a ligand-binding site in TRPC4. In addition, the authors suggest a new mechanism underlying the association of TRPC4 and calmodulin. Finally, Vinayagam et al. create a mechanistic model for positive and negative modulation of the TRPC4 channel. The study has the potential to have a strong impact on the field and provides new mechanistic insights into TRPC4 function. The reviewers found the manuscript to be well written and the data clearly presented; in general, reviewers are very enthusiastic about the structural data. However, the authors need to address a few critical points.

Essential revisions:

1) The main comment voiced by all three reviewers refers to the actual channel activation by the compound GFB-9289. From the calcium imaging data, it seems to be a weak activator. Given the fact that the structure in the presence of this activator is very similar to the closed, apo structures, the reviewers ask that functional studies be performed. The actual single-channel open probability attained in the presence of GFB-9289 should be presented. If single-channel recordings cannot be attained, macroscopic current recordings comparing the fraction of current activated in comparison with a full agonist like Englerin-A should be presented. These experiments are particularly important since these compounds (the GFB activator and the inhibitor) are relatively new and have not been fully characterized functionally. Similar electrophysiological experiments regarding the action of the inhibitor should be presented.

2) Throughout the manuscript, TRPC4 is discussed as a store-operated channel. This notion, however, is a highly controversial issue. There is a general consensus in the field that the well-known STIM/Orai interaction is responsible for most forms of store-operated calcium entry. The initial conformational coupling hypothesis stipulating a direct interaction of the IP3 receptor and the C-terminus of TRPC proteins has never been convincingly shown in living cells and has by and large been abandoned. A more critical and balanced discussion, in particular with regard to store-operated calcium entry, is absolutely required.

3) Also relating to the discussion of the data, the analysis and discussion of cryo-EM data and the gating model proposed, completely neglected to discuss the commonly accepted view that the physiologically relevant activation mechanisms of TRPC channels may rely on PLC-mediated production of DAG. The Introduction and Discussion sections should be edited accordingly. More importantly, the impact of the manuscript will be much stronger if the authors discussed how the DAG-induced opening of TRPC4 could potentially be integrated into the suggested 3D model of TRPC4.

4) TRPC4-CaM structure has a very weak density for CaM at 5A resolution. It is not clear how the authors are even able to assign a specific lobe of the CaM to these very poorly resolved densities. Moreover, it is also not clear how authors can provide atomic details for these interactions in Figure 5C if the resolution is at 5A.

[Editors' note: further revisions were suggested prior to acceptance, as described below.]

Thank you for resubmitting your work entitled "Structural basis of TRPC4 regulation by calmodulin and pharmacological agents" for further consideration by *eLife*. Your revised article has been evaluated by Olga Boudker (Senior Editor) and a Reviewing Editor.

The manuscript has been improved but there are some remaining issues that need to be addressed before acceptance, as outlined below:

We appreciate all the additional work done to answer reviewers' concerns, specially the two-electrode and patch-clamp recordings of the effects of the GFB compounds, which have revealed that all behave as inhibitors, highlighting the importance of detailed electrophysiological studies. Regarding the new records (Figure 1), it is of notice that the effect of the application of the GFB inhibitors seems to diminish during application. Please make clear if this is a real effect or if the compounds were applied during short periods and not during all the time indicated by the vertical dashed lines. It will be very useful if a horizontal bar indicating the exact time of application is placed in all records.

The form of the equation used to fit dose response data in Figure 1 is non-canonical and no definition of parameters is provided. The standard form of a binding isotherm or Hill equation is:

1/(1+(ic50/L)^n) , where ic50 is the 50% effect concentration, L is the concentration of the ligand and n is the Hill coefficient.

Please use this standard form and indicate the meaning and values of the fit parameters.

---

## [Author Response]

Essential revisions:1) The main comment voiced by all three reviewers refers to the actual channel activation by the compound GFB-9289. From the calcium imaging data, it seems to be a weak activator. Given the fact that the structure in the presence of this activator is very similar to the closed, apo structures, the reviewers ask that functional studies be performed. The actual single-channel open probability attained in the presence of GFB-9289 should be presented. If single-channel recordings cannot be attained, macroscopic current recordings comparing the fraction of current activated in comparison with a full agonist like Englerin-A should be presented. These experiments are particularly important since these compounds (the GFB activator and the inhibitor) are relatively new and have not been fully characterized functionally. Similar electrophysiological experiments regarding the action of the inhibitor should be presented.

We thank the reviewers for this very constructive suggestion. We performed two-electrodes measurements with *Xenopus oocytes* expressing TRPC4 from zebrafish. To our surprise, these patch clamp experiments demonstrated that both compounds, GFB-8438 and GFB-9289, did not act as activator but inhibited the activating effect of Englerin A (see Figure 1). As electrophysiological experiments are direct measurements of channel currents and thus activity, we believe that they are more conclusive compared to our previous fluorescence-based Ca^2+^ imaging data, which are more vulnerable to external influences. To further confirm that pyridazinone-based compounds can be categorized as TRPC4 inhibitors, we performed the same electrophysiological measurements with a third representative of this compound class, named GFB-8749. Indeed, an inhibitory effect was observed, showing a clear reduction of the activating effect of (-)-Englerin A. To ensure that GFB-8749 binds to the same binding pocket as the other two inhibitors, i.e. formed by helices S1-S4 within the VSLD, we determined the cryo-EM structure of TRPC4 in complex with GFB-8749 to a resolution of 3.8 Å. As anticipated, the structure confirmed that GFB-8749 resides in the same binding pocket. Thus, all three pyridazinone-based compounds act as inhibitors rather than activators. We have modified and rephrased our manuscript, and changed the figures, accordingly.

2) Throughout the manuscript, TRPC4 is discussed as a store-operated channel. This notion, however, is a highly controversial issue. There is a general consensus in the field that the well-known STIM/Orai interaction is responsible for most forms of store-operated calcium entry. The initial conformational coupling hypothesis stipulating a direct interaction of the IP3 receptor and the C-terminus of TRPC proteins has never been convincingly shown in living cells and has by and large been abandoned. A more critical and balanced discussion, in particular with regard to store-operated calcium entry, is absolutely required.

This is a good point. We have added the descriptions of TRPC4 as a store/receptor-operated channel to the Introduction citing the relevant literature.

3) Also relating to the discussion of the data, the analysis and discussion of cryo-EM data and the gating model proposed, completely neglected to discuss the commonly accepted view that the physiologically relevant activation mechanisms of TRPC channels may rely on PLC-mediated production of DAG. The Introduction and Discussion sections should be edited accordingly. More importantly, the impact of the manuscript will be much stronger if the authors discussed how the DAG-induced opening of TRPC4 could potentially be integrated into the suggested 3D model of TRPC4.

We have edited the Introduction and Discussion part accordingly and added DAG-induced opening of the channel as suggested.

4) TRPC4-CaM structure has a very weak density for CaM at 5A resolution. It is not clear how the authors are even able to assign a specific lobe of the CaM to these very poorly resolved densities. Moreover, it is also not clear how authors can provide atomic details for these interactions in Figure 5C if the resolution is at 5A.

The reviewer is right that this region of the structure, located at the periphery of the reconstruction, is less well resolved compared to the core region of TRPC4. Nevertheless, secondary structure elements in this region were clearly resolved, allowing us to unambiguously place one of the two lobes of CaM into the corresponding density using flexible fitting.

As both lobes, N- and C-terminal, are structurally almost identical on the backbone level (RMSD=1.27Å, Figure 5—figure supplements 6C) they can be equally well placed into the density. However, the C-terminal lobe gave us a slightly higher cross correlation (0.8102 vs. 0.8080 for the N-terminal lobe) and hence we proceeded with this one for the subsequent flexible fitting step into the density. Our placement was further validated by the local complementary interactions between side chains of TRPC4 and CaM, which are conserved between the N- and C-terminal lobe (Figure 5—figure supplements 6B and C).

To test the binding of the CaM lobes biochemically, we performed pull down experiments using the individual lobes of CaM as a bait to pull down purified zebrafish TRPC4. Expectedly, both lobes were able to pull down TRPC4 in a similar way (Figure 5—figure supplements 6A).

Since we used bi-lobed CaM for our structural analysis, it is therefore likely that our structure represents a mixture of N- and C- lobes.

We updated this part of the manuscript and revised the corresponding figure legends (Figure 5, Figure 5—figure supplements 6).

[Editors' note: further revisions were suggested prior to acceptance, as described below.]

The manuscript has been improved but there are some remaining issues that need to be addressed before acceptance, as outlined below:We appreciate all the additional work done to answer reviewers' concerns, specially the two-electrode and patch-clamp recordings of the effects of the GFB compounds, which have revealed that all behave as inhibitors, highlighting the importance of detailed electrophysiological studies. Regarding the new records (Figure 1), it is of notice that the effect of the application of the GFB inhibitors seems to diminish during application. Please make clear if this is a real effect or if the compounds were applied during short periods and not during all the time indicated by the vertical dashed lines. It will be very useful if a horizontal bar indicating the exact time of application is placed in all records.

It is indeed true that the effect of the GFB inhibitors, especially GFB-9289, diminished during application. We applied GFB inhibitors between the dashed lines (not only at the dashed lines). We relabelled Figure 1 A, E, I, B, F, J, D, H, L with colored horizontal bars and describe it in the revised figure legend.

The form of the equation used to fit dose response data in Figure 1 is non-canonical and no definition of parameters is provided. The standard form of a binding isotherm or Hill equation is:1/(1+(ic50/L)^n) , where ic50 is the 50% effect concentration, L is the concentration of the ligand and n is the Hill coefficient.Please use this standard form and indicate the meaning and values of the fit parameters.

As proposed, we used now the Hill equation to fit the dose-response curves in Figure 1C, G, K and described the applied parameters in the legend of C, G, K as “(C, G and K) Hill equation: y = 1/ (1+ (IC_50_/x) ^ n) was used to fit the dose-inhibition curve, where IC_50_ is the 50% inhibitory concentration, x is the concentration of the ligand and n is the Hill coefficient.”